# Electric field dynamics in the brain during multi-electrode transcranial electric stimulation

Ivan Alekseichuk [1], Arnaud Y. Falchier[2], Gary Linn[2], Ting Xu[3], Michael P. Milham[2,3],
Charles E. Schroeder[2,4] & Alexander Opitz[1]

Neural oscillations play a crucial role in communication between remote brain areas. Transcranial electric stimulation with alternating currents (TACS) can manipulate these brain oscillations in a non-invasive manner. Recently, TACS using multiple electrodes with phase shifted stimulation currents were developed to alter long-range connectivity. Typically, an increase in coordination between two areas is assumed when they experience an in-phase stimulation and a disorganization through an anti-phase stimulation. However, the underlying biophysics of multi-electrode TACS has not been studied in detail. Here, we leverage direct invasive recordings from two non-human primates during multi-electrode TACS to characterize electric field magnitude and phase as a function of the phase of stimulation currents. Further, we report a novel "traveling wave" stimulation where the location of the electric field maximum changes over the stimulation cycle. Our results provide a mechanistic understanding of the biophysics of multi-electrode TACS and enable future developments of novel stimulation protocols.

[1] Department of Biomedical Engineering, University of Minnesota, Minneapolis 55455 MN, USA. [2] Center for Biomedical Imaging and Neuromodulation, Nathan Kline Institute for Psychiatric Research, Orangeburg 10962 NY, USA. [3] Center for the Developing Brain, Child Mind Institute, New York 10022 NY, USA. [4] Departments of Neurological Surgery and Psychiatry, Columbia University College of Physicians and Surgeons, New York 10032 NY, USA. Correspondence and requests for materials should be addressed to A.O. (email: aopitz@umn.edu)

Efficient brain function relies on a sophisticated multi-scale system of neuronal communication. Neural communication at the level of synaptic signaling gives rise to synchronized excitability fluctuations in neuronal ensembles at the regional and network levels, which in turn emerge as global brain oscillations or rhythms[1]. Brain rhythms reflect neural operations underlying behavior and cognition[2–4] and are present in all mammals across their evolution[5]. Their abnormality is a significant factor in psychiatric disorders, such as schizophrenia, ADHD, depression, and anxiety[6,7]. The rising appeal and evident importance of neural oscillations[8] that both reflect[1] and shape[9] neural communications inspire the search for tools allowing their causal manipulation.

Recent developments enable such manipulations using weak transcranially applied electric fields. Transcranial alternating current stimulation (TACS) can modulate brain rhythms by coupling intrinsic electric fields in the brain to externally applied electric currents in a matching frequency band[10–12]. Such coupling can magnify the power or realign the phase of ongoing brain rhythms. Thus, TACS presents an exciting tool to causally probe the physiological and behavioral role of brain rhythms and their synchronization, i.e., connectivity[13,14]. Several studies have used TACS to manipulate within- and between-area brain connectivity[15–18]. The latter requires a departure from conventional two-electrode stimulation techniques and the use of model-driven, multi-electrode approaches to drive two or more brain sites independently. Through the simultaneous entrainment to the rhythm of external electric fields, researchers can manipulate the phase co-alignment of intrinsic oscillations and study their functional importance.

Modulation of between-area brain connectivity first was done via dual-site TACS where two stimulation electrodes cover two target brain regions and a third electrode serves as a return electrode outside the regions of interest. Following a seminal experimental study[15], multiple groups employed this procedure[18–21] or modifications of it[22,23]. So far, all studies have considered two primary stimulation conditions: either the phase of the alternating current was the same (0° shift) between the two stimulation electrodes or it was the opposite (180° shift). Here, the core idea is that neural oscillations in the target brain areas will resonate with the applied electric currents and align their phases accordingly. Because phase alignment of oscillations in communicating brain regions is functionally important[24–27], application of in-phase or out-of-phase currents should facilitate or hinder their communication. This principle has guided the design and interpretation of these dual-site and, in general, multi-electrode TACS experiments. However, recent modeling efforts dispute their biophysical validity with regard to the achieved phase differences of stimulation fields[28].

The current rationale for multi-electrode TACS applications relies on two main assumptions. The first is that the manipulation of phase differences between stimulation electrodes does not significantly change other properties of the generated electric field. Most importantly, this predicts the field's spatial configuration, i.e., the electric field magnitude at different spatial locations. However, direct measurements to substantiate this assumption are lacking. The second assumption is that the phase of the electric field in the brain underneath the stimulation electrodes is the same as the phase of the currents passing through these electrodes. This assumption holds for the two electrode case[29], where the only possible obstacle for such translation could be capacitive effects introducing additional phase shifts. However, fundamental electromagnetic theory predicts based on the superposition principle that a spatially varying phase gradient in the electric field can arise under certain conditions from multi-electrode TACS (see Supplementary

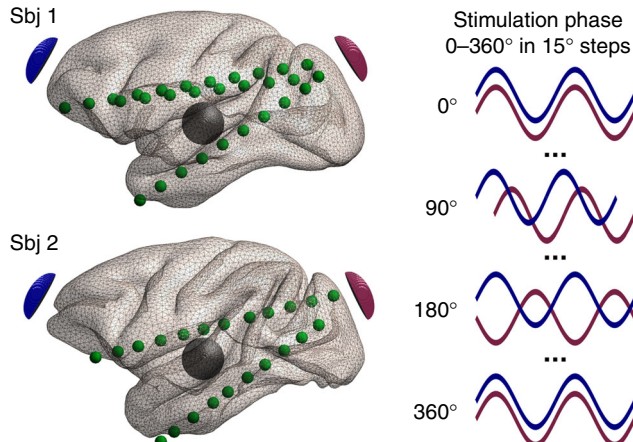

**Fig. 1** Experimental design. Two nonhuman primates with recording electrodes implanted along the anterior-posterior plane receive transcranial electric stimulation (TES). For subject 1 we analyze three recording electrodes with 29 contacts altogether, and for subject 2—two recording electrodes with 22 contacts. Two TES stimulation electrodes are placed on the scalp at anterior (blue) and posterior (red) locations, with the return electrode (black) over the temporal area. Alternating current stimulation was applied at 10 Hz and fixed intensity at the stimulation electrodes, while the phase of the stimulation currents between them varied from 0° to 360° in 15° steps (25 phase conditions in total). See Supplementary Movie 1 for a 3D animation

Discussion). While such behavior can be expected theoretically, it has not been demonstrated experimentally using in vivo recordings in the brain.

Here, we aim to address this gap by measuring electric field magnitudes and phase angles across the brain during three-electrode TACS under varying phase conditions. Nonhuman primate models have proven themselves as useful to study biophysical and physiological TES mechanisms[29–33] as their brains more closely resemble human brain anatomy than do the brains of nonprimate species. We applied multi-electrode TACS and characterized the spatiotemporal properties of generated electric fields, performing direct, stereotactic EEG measurements (Fig. 1 and Supplementary Movie 1), while varying the phase differences between stimulation currents from 0° to 360° in 15° steps. We show in vivo a nonlinear relationship between the phase difference of transcranially applied currents and the phases and magnitude of measured intracranial electric fields. We further describe a previously unreported capability of multi-electrode TACS to generate "traveling wave" electric fields in the brain, enabling the design of novel stimulation protocols.

## Results

**Magnitude and phase of the electric field**. To assess the voltage and electric field distributions in the brain arising from multi-phase TACS, we conducted direct, intracranial measurements in two nonhuman primates (Fig. 1 and Supplementary Movie 1). The voltage distribution in the brain demonstrates a clear dependence of the stimulation phase (Fig. 2a, b). A much smaller range of TES voltages in the brain was found for the 0° condition with 6.64 mV for the subject 1 and 15.19 mV for the subject 2, compared to the 180° condition with 30.18 and 43.77 mV, respectively. One-way (factor: applied stimulation phase) ANOVA of deviations from the median confirms a strong disparity in voltages across the stimulation conditions (subject 1: $F_{24,700} = 15.54$, $p = 3.3 \times 10^{-50}$; subject 2: $F_{24,525} = 7.45$, $p = 1.6 \times 10^{-21}$). The TES voltage gradient is oriented along the anterior–posterior plane with its minimum and maximum

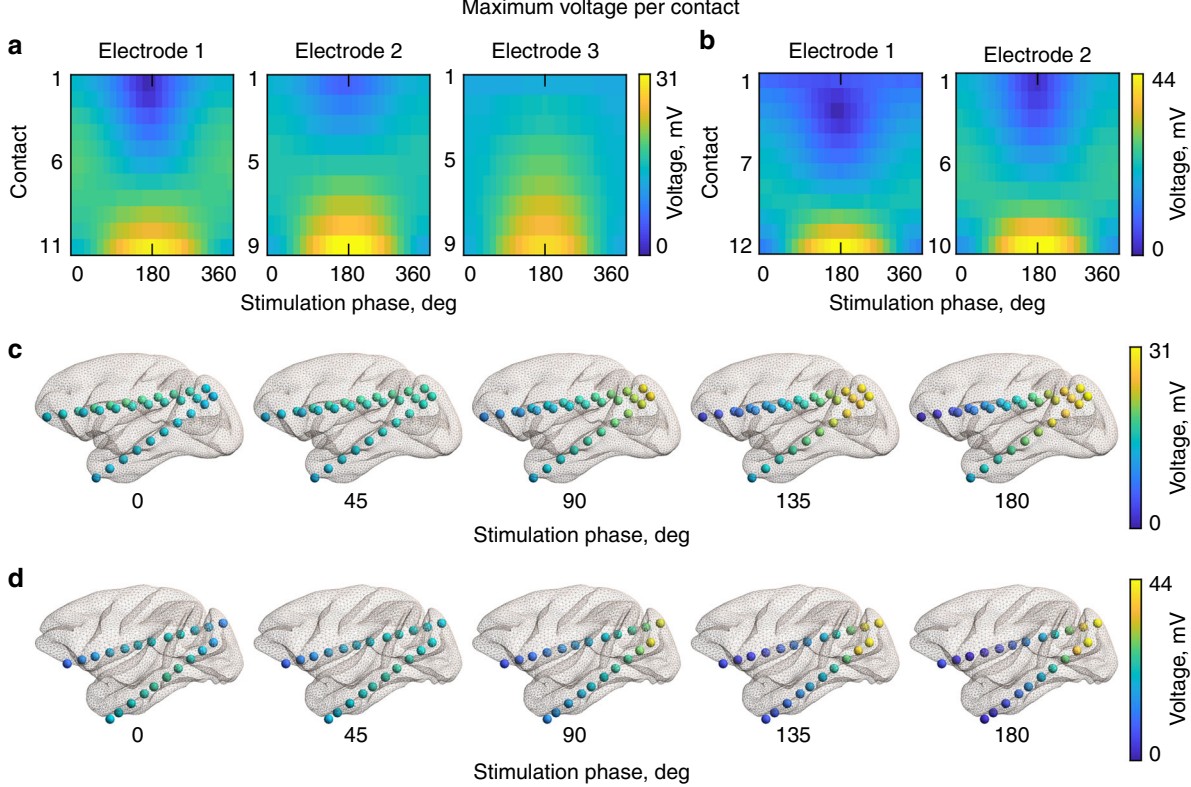

**Fig. 2** Voltage distribution during TES for different stimulation conditions. **a** Heatmaps of the maximum voltage for each recording electrode for subject 1 (left) and subject 2 (right, **b**). The x-axis indicates the applied phase difference between the anterior and posterior stimulation electrodes from 0° to 360° in 15° steps, and the y-axis corresponds to the recording contacts (from the first—most anterior contact, to the last—most posterior contact). **c** Voltage gradients for select stimulation conditions for subject 1 and subject 2 (**d**) visualized at their recorded anatomical locations

occurring in the proximity to the target stimulation electrodes (Fig. 2c, d).

We found a similar picture for the magnitude of the electric field (Fig. 3a, b; and Supplementary Fig. 4). The maximum electric field is significantly weaker for the 0° than for 180° condition (nonparametric one-way ANOVA for the subject 1: $\chi^2_{24,700} = 396.2$, $p = 4.5 \times 10^{-69}$; for the subject 2: $\chi^2_{24,525} = 164.3$, $p = 6.95 \times 10^{-23}$). Relatively higher electric field magnitudes in posterior brain regions are due to their closer proximity to the stimulation electrode compared to anterior brain regions. Further, we found the electric field strength to be higher in superficial brain regions for the 0° condition and higher magnitudes at progressively deeper brain regions with increasing stimulation phase difference up to 180° (Fig. 3c, d). The relative ratio of the electric field strength at the most superficial contact compared to the deepest recording contact were found as follows: subject 1 at 0° condition = 11.35, at 45° = 2.59, at 90° = 2.09, at 180° = 1.93; for subject 2 at 0° = 10.95, at 45° = 3.32, at 90° = 2.79, at 180° = 2.5.

Further investigating the stimulation phase dependency of TES voltages and electric fields, we observed a nonlinear increase from 0° to 180° and a nonlinear decrease from 180° to 360°. We found that a sinusoidal curve fit the voltage data very well, SSE (sum of squares due to error) = 1.29 and $R^2_{adj} = 0.99$ for the subject 1, and SSE = 4.25 and $R^2_{adj} = 0.99$ for the subject 2. For comparison, linear fit exhibits an order of magnitude larger errors (for the subject 1: SSE = 29.9, $R^2_{adj} = 0.91$; for the subject 2: SSE = 71.45, $R^2_{adj} = 0.91$). A similar relationship was observed for the magnitude of the electric field. A sinusoidal curve fit the data almost perfectly (for subject 1: SSE = 0.001, $R^2_{adj} = 0.99$; for subject 2: SSE = 0.001, $R^2_{adj} = 0.99$), though the linear fit is nearly

as good (for subject 1: SSE = 0.01, $R^2_{adj} = 0.96$; for subject 2: SSE = 0.04, $R^2_{adj} = 0.95$).

Next, we analyzed how the phase angles of voltage ($\varphi_V$) and electric field ($\varphi_E$) in the brain depend on the stimulation conditions. The conditions strongly affect $\varphi_V$ (One-way ANOVA for circular data for subject 1: $F_{24,700} = 5.23$, $p = 3.03 \times 10^{-14}$; subject 2: $F_{24,525} = 6.55$, $p < 1 \times 10^{-15}$) and $\varphi_E$ (subject 1: $F_{24,700} = 97.6$, $p < 1 \times 10^{-15}$; subject 2: $F_{24,525} = 6.61$, $p < 1 \times 10^{-15}$) although in different ways. To analyze how large phase differences can occur across the brain, we computed the phase difference between the most anterior and most posterior recording contact for each electrode. For the voltage phase ($\Delta\varphi_V$), we found close to zero differences for both 0° (equivalent to 360°) and 180° stimulation conditions (Fig. 4a, c, d) as previously reported[29]. $\Delta\varphi_V$ is maximal for 135°–150° conditions and, symmetrically, for 210°–225° conditions. At the same time, the anterior–posterior difference in phase angles of the electric field ($\Delta\varphi_E$) is maximal for 0°/360° conditions and close to zero for 180° condition (Fig. 4b). Once again, the changes are not linear across conditions and are better approximated by a sinusoidal curve (all $R^2_{adj} \geq 0.98$). Considering the spatial distribution of $\Delta\varphi_E$ (Fig. 4e, f; and Supplementary Figs. 5–7), the zero-phase difference during 180° stimulation condition indicates a unidirectional electric field along the recording electrodes. Large $\Delta\varphi_E$ characteristic for the 0° stimulation condition is a sign of bidirectional electric fields, such as inward- or outward-oriented.

**Traveling wave stimulation.** In further analysis, we investigated the evolution of electric fields over time. Electric fields in the brain vary over time with the same period as the applied current (Supplementary Fig. 3). Figure 5 shows the temporal evolution of

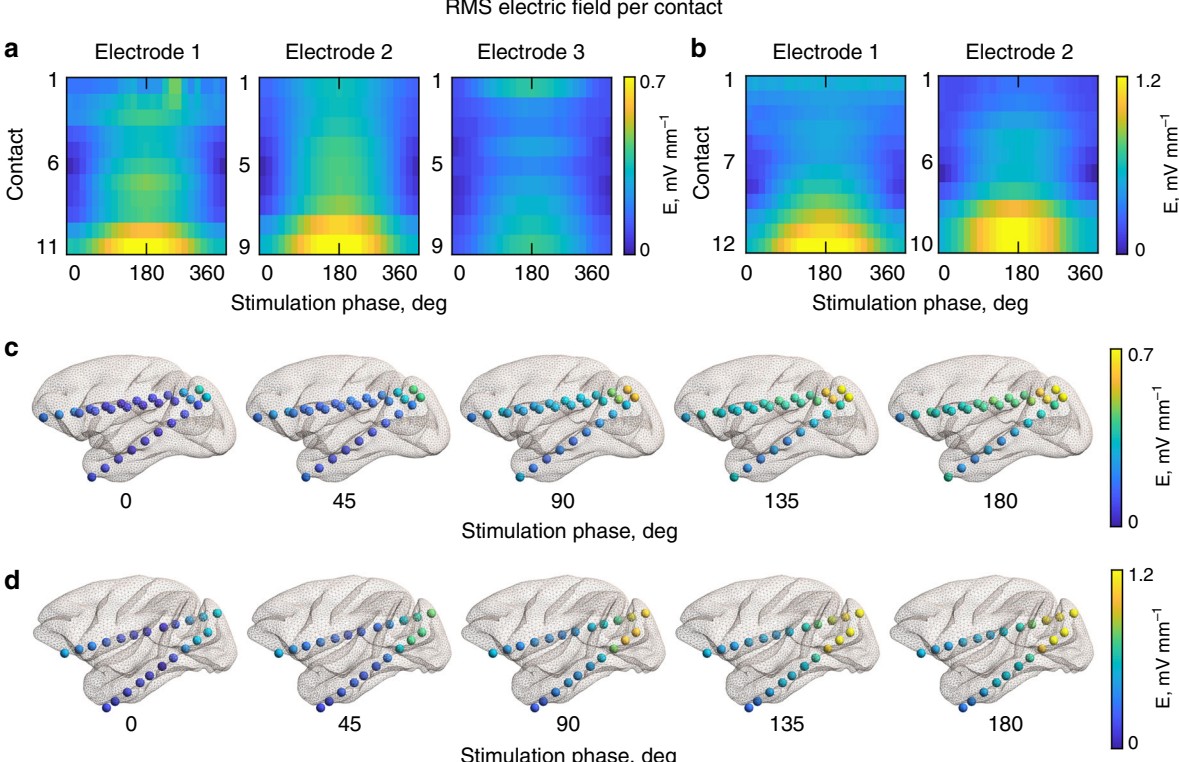

**Fig. 3** TES electric field distribution for different stimulation conditions. **a** Heatmaps of the root mean square magnitude (RMS) of the electric field for each recording electrode for subject 1 (left) and subject 2 (right, **b**). The x-axis indicates the applied phase difference between the anterior and posterior stimulation electrodes from 0° to 360° in 15° steps, and the y-axis corresponds to the recording contacts (from the first—most anterior contact, to the last —most posterior contact). **c** Electric field distributions in the brain for select stimulation conditions for subject 1 and subject 2 (**d**). See also Supplementary Fig. 4 for further details

electric fields for three main stimulation conditions (0°, 90°, and 180°; see also Supplementary Movies 2 and 3). Stimulation in the 180° condition leads to a unidirectional electric field for all time-points. The electric field direction is periodically switching from an anterior- > posterior to a posterior- > anterior orientation. Stimulation in the 0° condition has a reverse effect: the electric field is bidirectional between the anterior and posterior brain region for all time-points while the direction is changing between inward and outward orientations. Stimulation in the 90° condition creates an intermediate scenario with the electric field being unidirectional for 50% of the time, inward-oriented for 25% and outward-oriented in another 25% of the time. The spatial location of the maximum of the electric field strength was found to be at the same location for the 0° and 180° stimulation conditions for all time points. Intriguingly, intermediate stimulation conditions, such as the 90° condition, can generate a "traveling wave" in the location of the electric field maximum. This means that the spatial location of the stimulation maximum at a given time point varies over the stimulation period (Supplementary Movie 4).

In an additional analysis we further quantified this "traveling wave" phenomenon in more detail. For this we normalized the electric field time-courses to the individual maximum for each stimulation condition and each recording contact. This was done such that different electric field distributions can be easily compared across all stimulation conditions and locations. We defined a dissimilarity index as the mean squared difference of the normalized electric fields across time points. A dissimilarity index score closes to zero was found for the 0° and 180° stimulation conditions, indicating that for each contact the electric field maximum occurred at the same time point (Fig. 6a, b). However, other stimulation conditions, most noticeably the 45° condition

and symmetric to it the 315° condition, are characterized by a strong dissimilarity of electric field time-courses. This means that the spatial locations of their electric field maxima are traveling over time. Further, the spatial location of these electric field maxima showed a clear temporal pattern (Fig. 6c and Supplementary Movie 4; see Supplementary Fig. 9 for corresponding nonnormalized data). We found that the relative maximum of the electric field is periodically and gradually moving between the most anterior and the most posterior contact, thus creating a "traveling wave" of electric stimulation.

Mathematically, "traveling wave" can be described with a spatial phase gradient. Using linear regression of the electric field phases $\varphi_E$ along the recording electrode contacts we can estimate the spatial frequency $r$ and propagation speed $c$ of the traveling wave[34]. We found the median spatial frequency $r$ for subject 1 to be 1.42 deg mm$^{-1}$ (range: 0.06–4.8 deg mm$^{-1}$), 1.28 deg mm$^{-1}$ (0.08–5.88 deg mm$^{-1}$), and 1.06 deg mm$^{-1}$ (0.01–5.8 deg mm$^{-1}$) along electrodes 1–3 across phase conditions. The median spatial frequency $r$ for subject 2 is 1.5 deg mm$^{-1}$ (0–4.25 deg mm$^{-1}$) along electrode 1 and 1.49 deg mm$^{-1}$ (0.07–5.57 deg mm$^{-1}$) along electrode 2. The spatial frequency is almost zero for stimulation with 180° ± 60° current phase shift. We interpret $r \geq$ 1 deg mm$^{-1}$ as a sufficient gradient to detect a traveling wave[34,35]. This analysis agrees with the dissimilarity index (Fig. 6a, b). The median propagation speed $c$ of the traveling waves is 1.64 mm ms$^{-1}$ (range: 0.75–3.99 mm ms$^{-1}$), 1.63 mm ms$^{-1}$ (0.61–4.19 mm ms$^{-1}$), and 2 mm ms$^{-1}$ (0.62–5.74 mm ms$^{-1}$) along electrodes 1–3 in subject 1; and 1.57 mm ms$^{-1}$ (range: 0.85–3.82 mm ms$^{-1}$) and 1.47 mm ms$^{-1}$ (0.65–4.1 mm ms$^{-1}$) along electrodes 1 an 2 in subject 2.

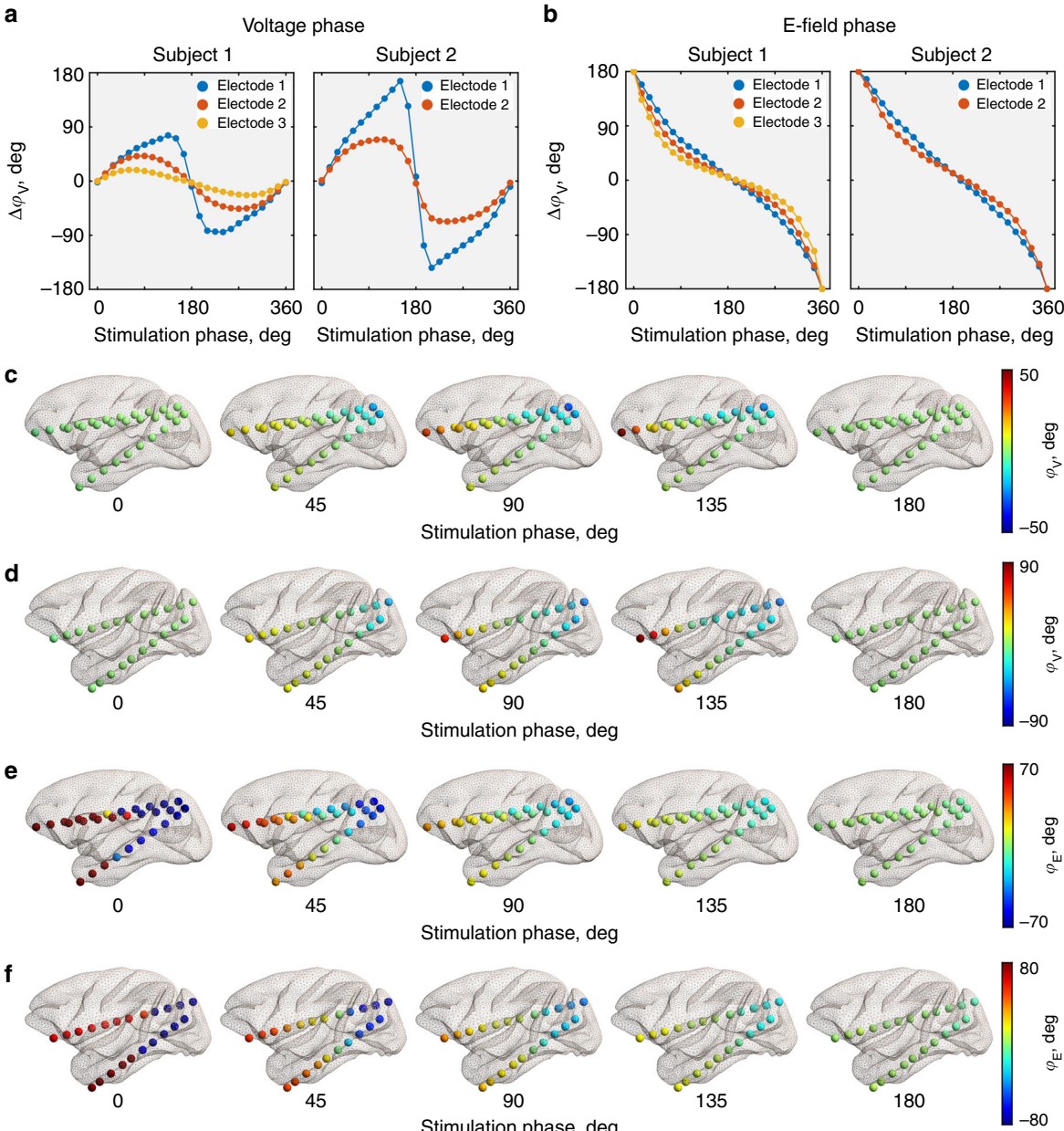

**Fig. 4** Voltage and electric field phases in the brain during TES for a given stimulation condition. **a** The difference between the voltage phases recorded from the most anterior and most posterior contact ($\Delta\varphi_V$) per each electrode. The left figure corresponds to subject 1, and the right figure to subject 2. **b** Same as panel a, but for the electric field phase differences ($\Delta\varphi_E$). **c** 3D visualizations of the voltage phases ($\varphi_V$) in the brain for the primary stimulation conditions at all recording contacts for subject 1 and subject 2 (**d**). **e**, **f** Same as panels **c** and **d**, but for the electric field phase ($\varphi_E$). See also Supplementary Figs. 5–7 for further details

To establish the generalizability of the "traveling wave" stimulation, we conducted a control experiment. For subject 2, we rotated the stimulation electrode-montage: one active electrode was located on the left temple, another—on the forehead, and the return electrode—on the back of the head (Supplementary Fig. 10a). Thus, the generated electric field was perpendicular to the one in the main experiment. The phenomenon of a "traveling wave" remained present (Supplementary Fig. 10b–d), although the most optimal phase shift of stimulation currents for its generation was also rotated and maximized at 165° and 195°. Considering future applications of "traveling wave" TACS, adequate modeling will be necessary to establish optimal parameters for a given configuration of the stimulation electrodes.

To further explore the generalizability of our findings, we performed a computer simulation of the main experimental conditions using a realistic finite element method (FEM) model of the monkey brain (Fig. 7). Both 0° and 180° stimulation conditions resulted in electric fields which have stable spatial distributions throughout the time-course. However, the "traveling wave" (45°) condition resulted in an electric field which gradually moved from the posterior to the anterior neocortex during every half-cycle of the stimulation (see Supplementary Movies 5–7 for normalized and directed electric fields). Interestingly, despite a twice stronger current in the 0° than in the 180° stimulation condition, all models indicate the same maximum electric field in the gray matter of 2.4 mV mm$^{-1}$ (per 1 mA on both active electrodes). This is

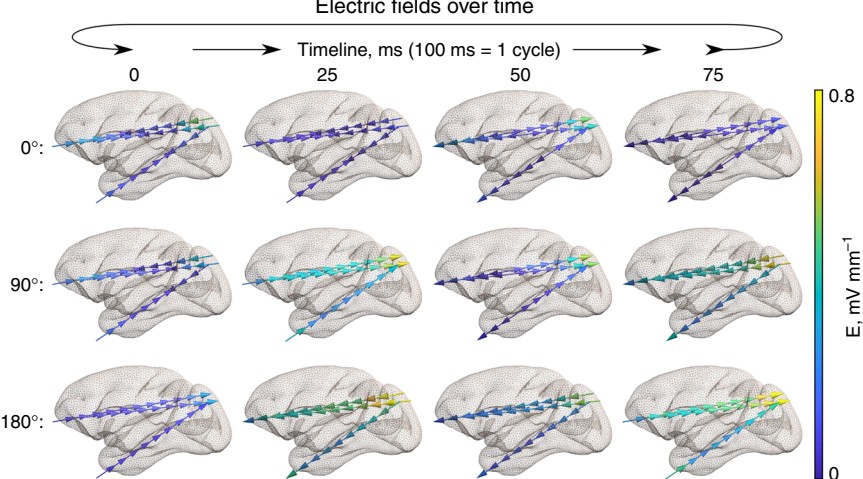

**Fig. 5** Electric fields in the brain over time during TES for a given stimulation condition. The panel depicts the main conditions (0°, 90°, and 180° stimulation phase differences) for subject 1. Arrows indicate the electric field direction, and the color encodes the electric field magnitude. See Supplementary Fig. 9 for subject 2 and Supplementary Movies 2 and 3 for more details

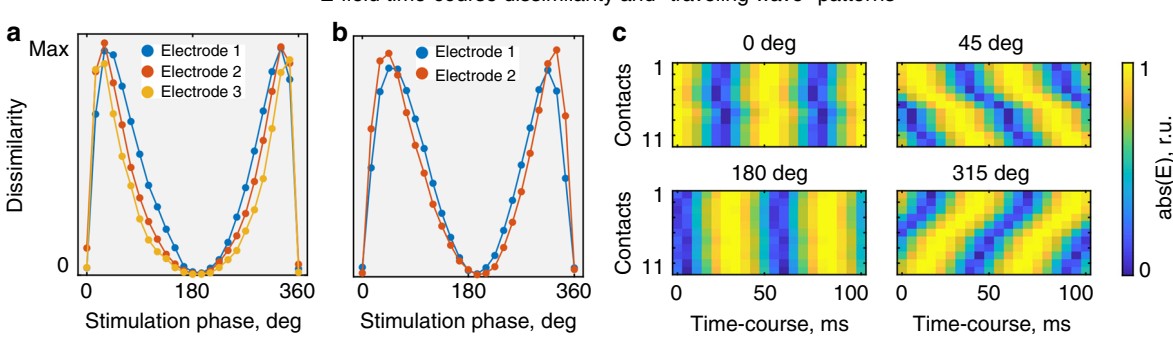

**Fig. 6** Traveling wave stimulation. Electric field time-courses across different recording contacts ( = anatomical locations) are identical for 0°/360° and 180° stimulation conditions but demonstrate traveling wave properties for intermediate stimulation conditions, e.g., 45°. **a**, **b** Dissimilarity (mean square difference) between the electric field time-courses at different contacts. The left figure corresponds to subject 1, and the right figure to subject 2. **c** Absolute normalized (per location) electric field time-courses for subject 1 and electrode 1 in the relative units (r.u.). The first contact in the electrode corresponds to the most anterior location, and the last contact—to the most posterior location. While for 0° and 180° the maxima across contacts occur at the same time point, they occur at different time points for the 45° condition ( = traveling wave). Other electrodes demonstrate a similar pattern. See Supplementary Fig. 9 for corresponding nonnormalized data and Supplementary Movie 4 for a 3D animation

likely due to the higher percentage of shunted current in the otherwise stronger 0° condition, as the involved electrodes on the scalp are closer to each other. To quantify the stimulation results, we rescaled the electric fields between zero and one for each time-step and estimated the standard deviation at every location of the gray matter across time. Its maximum was 0.27 for the 45° stimulation condition, showing high spatial variability across time. Maximum standard deviation was found to be 0 for both 0° and 180° stimulation conditions, indicating no spatial change over time.

Mathematically, the electric field **E** (including magnitude and phase) arising from phase-shifted TES can be calculated for any brain region leveraging the principles of phasor analysis (see Supplementary Discussion for the mathematical derivation). Performing this analysis shows smooth gradual changes of the electric field phase across the neocortex for stimulation conditions such as 45°, but not for 0° or 180° (Supplementary Fig. 11). Thus, our modeling results and analytical considerations can be used to describe the experimental recordings and also demonstrate "traveling" fields during the 45° but not 0° or 180° stimulation conditions.

## Discussion

The present work quantified the biophysical features of multi-electrode, multi-phase TES, which forms the foundation for explaining and predicting its neuromodulatory effects. We directly recorded and analyzed the electric potential and field in the nonhuman primate brain arising from three-electrode transcranial alternating current stimulation. Our study has three main findings: (i) differing electric field magnitude across stimulation phase conditions; (ii) nonlinear relationship between transcranial stimulation phase and measured intracranial phase; (iii) specific phase configurations can create traveling wave stimulation patterns. Importantly, all measurements were repeated in two nonhuman primates from two taxonomic families with substantially different head anatomy. Our clear and highly comparable results encourage the generalization and translation of the present findings to all primates, including humans.

The dominant method of dual-site TACS in human studies is the application of a three-electrode montage where two stimulation electrodes are either in-phase (0° condition) or in anti-phase (180° condition) with each other[15,18–21]. One key assumption behind this approach is that the generated electric field is comparable in any

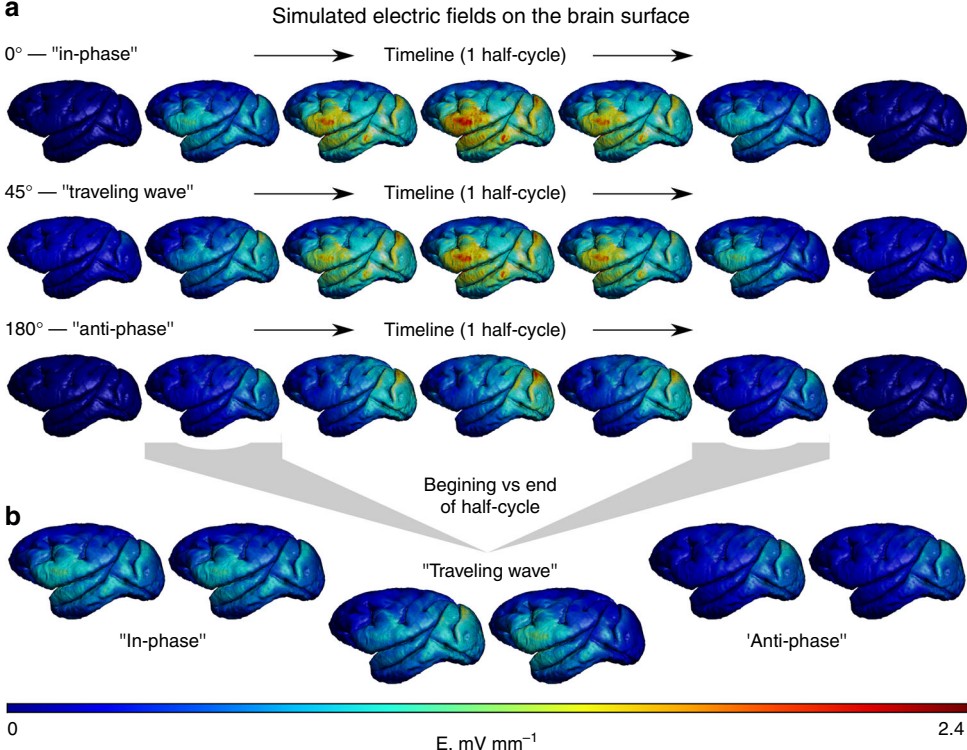

**Fig. 7** Computer model of the electric field on the gray matter surface during conventional stimulation with 0° and 180° phase difference and during "traveling wave" stimulation (45° phase difference). Panel **a** depicts the changes throughout the time (one half cycle of stimulation), and panel **b** highlights two time-points at the begging and end of half-cycle. See Supplementary Movies 5–7 and Supplementary Fig. 11 for more details

aspect but phase. However, recent modeling work challenged this assumption and demonstrated that electric fields differ significantly between the two conditions[28]. Our experimental data support and extend these model-driven considerations. We found that the electric field strength is significant larger (2–2.3 times) for the anti-phase than for the in-phase stimulation condition. This difference in electric field strength should thus be revisited as a potential driver of the behavioral or physiological effects reported between in-phase or anti-phase stimulation in the literature. This will also apply to four- or more electrode montages that prescribe a single stimulation electrode per each brain area of interest[22,23]. However, further research is needed to investigate Laplacian montages where multiple electrodes are positioned around a single brain area of interest for maximal engagement[36–38]. Importantly, the electric field magnitude between stimulation conditions is changing in a nonlinear manner which is well approximated by a sinusoidal curve. Here, we should note that the measured electric fields are restricted to their anterior-posterior component along the implanted electrodes. It is to be expected that in the 0° conditions medial-lateral components will be larger than for the 180° condition. This does not diminish the validity of our findings as the direction of the electric field with respect to neural elements is most important for TES physiological effects[39,40]. The electric fields with different directions will affect a different subset of neurons in the brain. Thus, differences in one electric field component across stimulation conditions are important to document and control for. Here, our measurements can inform future experimental work so that it can better control for these effects. Moreover, we performed electric field modeling and phasor analysis of the main experimental conditions. These models describe all directions of the electric fields in the brain and are in full agreement with the experimental results. This further strengthens the generalizability of our findings.

Besides the electric field magnitude, we analyzed the electric field phase as a function of stimulation condition. Contrary to the intuitive expectation, the in-phase (0°) stimulation condition resulted in an "anti-phase" electric field, where anterior and posterior brain areas experience bidirectional electric fields at any moment in time. The spatial location of the deflection point (from anterior oriented to posterior oriented electric fields) crucially depends on the location of the return electrode and future studies can aim to manipulate this by carefully choosing the location of stimulation and return electrodes. The anti-phase (180°) condition leads to an "in-phase" or unidirectional electric field for all contacts and all time-points. Here, the electric field direction is flipping over time between anterior- and posterior-directed states. These differences in electric field directionality might be a driving force for previous findings. Of course, this is complicated by the fact of differing neuronal orientations with respect to the TES electric field in a folded cortex[41].

The picture for intermediate phase conditions, such as 90° or 270°, is more intricate. It is a mixture of bi- and uni-directional electric field states that reoccur over time. Overall, the complexity of the relationship between phase conditions in multi-electrode TES and electric field phase presents a challenge and an opportunity for future experimental and modeling studies. Precise measurements are especially important for efforts regarding TES artifact rejections due their interaction with physiological signals[42]. Our experimental data underscore the point that explicitly accounting for the full temporal dynamics of electric fields can significantly expand the understanding of TES effects, especially for multi-electrode stimulation.

One novel intriguing finding that arises from the temporal analysis of TES electric fields across the whole spectrum of phase conditions is a "traveling wave" stimulation condition. It is characterized by a gradual and periodic movement of the electric field maximum in the brain during multi-electrode, multi-phase alternating current stimulation. Such propagation of electric fields resembles the well-known electrophysiological phenomenon of traveling waves in the brain[35,43,44]. At the macroscopic level, these traveling waves of electrical activity contribute to the

formation of large-scale cell assembles and neural networks[34]. Traveling waves are observed and functionally relevant in different frequency bands, brain states and anatomical regions, including theta activity in the hippocampus[45], beta activity in the motor cortex[46], and gamma activity in the visual cortex[47]. Traveling waves can even occur on a global level spanning the whole neocortex[48,49]. Recently, it was shown that the consistency and power of large-scale traveling waves in the human neocortex are reliably correlated with the performance of an ongoing memory task[34], visual perception[50], and with memory consolidation during sleep[51]. The properties of these phenomena vary with respect to the underlying neurophysiological process or task and brain state. In the human brain, traveling wave propagation speed ranges from 0.25 to 10 mm ms$^{-1}$, and its spatial extent varies from few millimeters to dozens of centimeters[34,35,50]. In our experiments, we generated traveling waves propagating at speeds of 1.5–2 mm ms$^{-1}$. However, multi-electrode TES can directly control the speed and the phase gradient of the electric fields through the choice of stimulation parameters (e.g., frequency, phase, and electrode montage). Traveling wave TES thus presents a novel tool to mimic and manipulate such naturally occurring brain activity, obtain causal rather than correlational evidence for its role, and eventually converge into a novel therapeutic approach for cognitive deficits and neuropsychiatric disorders.

The present study confirms and expands previous work on the biophysical mechanisms of TES in nonhuman primates[29,30] and epilepsy patients[52]. Previous studies conducted for classic two-electrode montages found that TACS generated electric fields behave in an ohmic manner with minor-to-negligible phase shifts in the low frequency range. We can thus generalize our results to other low frequencies in the EEG range as well. Using phase-shifted input currents from a three-electrode montage, we demonstrate the possibility of phase differences of TES voltages and electric fields. This opens the possibility to create stimulation protocols to stimulate remote brain regions at different phase relations. While this was intended in previous studies[15,18,19,21–23], they were not based on biophysical considerations of TES electric fields. Both for the purpose of TES modeling and EEG source analysis, the head is assumed as a quasi-static volume conductor, because in the frequency range of interest (<1000 Hz) capacitive components of tissue impedance, inductive effects, and electromagnetic propagation effects are negligibly small[53]. However, through the combination of multiple sources with different phases, it is possible to control the local phase and create phase gradients across remote brain regions. The exact features of the electric field will depend both on the stimulation current phases and on the electrode placement, as we show through the additional experiment. Notably, we demonstrate that modeling software such as SimNIBS[54] can be used to generate realistic simulations of traveling wave protocols. Future work that includes new optimization schemes will allow a more principled design of multi-phase stimulation protocols.

In summary, the findings of the present work address important biophysical aspects of multielectrode TES. Previous studies investigated the effect of electric fields on the neurophysiology using various experimental approaches from in silico[55] to in vitro[56,57] and in vivo[58]. Seminal work from Terzolo and Bullock (1956) demonstrated the sensitivity of neurons to weak electric fields (≈1 mV mm$^{-1}$) strongly depending on the electric field orientation with respect to the somatodendritic axis[59]. More recent studies show the importance of cellular and network level states, and their coherence with the stimulation parameters[55–58]. However, surprisingly little is known about the electric fields that TES produces in the brain. System level effects of TES are driven by physical factors such as "where" and "when" the generated electric fields interact with neural tissue. While the "where" question has been explored in previous studies[60–62], the "when" question

specific for alternating current stimulation has been largely missing so far. Here, we examined the spatiotemporal properties of multi-electrode TACS with direct, stereotactic recordings in the nonhuman primate brain. The results show the importance of the stimulation phase both for the phase and magnitude of the generated electric field. Further, we demonstrated that multi-electrode, multi-phase TES can create a "traveling wave" stimulation where the location of the maximum stimulation changes over time. The present study opens new possibilities for exploiting the temporal complexity of multi-electrode, multi-phase TES and enables future efforts to tie together the "when", "where", and "how" of TES into a comprehensive, predictive model.

## Methods

**Subjects**. Experiments were conducted in two nonhuman primates. All procedures were approved by the Institutional Animal Care and Use Committee of the Nathan Kline Institute for Psychiatric Research. Subject 1 is a capuchin monkey (*Cebus apella*, 11 y.o., female, 2.9 kg) and subject 2 is a rhesus monkey (*Macaca mulatta*, 6 y.o., female, 4.8 kg). Both subjects were implanted with MRI-compatible head posts and three multicontact stereo-EEG depth electrodes (Ad-Tech) that sampled the intracranial electric field at 5 mm intervals in the posterior-anterior direction. The electrodes were implanted via a small craniotomy over the left occipital cortex with the terminal points in the frontal eye field (12 contacts, 5 mm spacing), medial prefrontal cortex (10 contacts, 5 mm spacing) and anterior hippocampus (10 contacts, 5 mm spacing). The craniotomy was sealed after the injection with nonconducting bone cement. The electrode locations were confirmed by post-implantation magnetic resonance imaging.

**Transcranial alternating current stimulation**. TACS was delivered using a current-controlled, multi-electrode system StarStim (Neuroelectrics™) via three round Ag/AgCl electrodes (Pistim™, radius = 3.14 cm$^2$) with conductive gel (SignaGel™). The electrodes were placed on the scalp over the middle forehead (first stimulation electrode), the left occipital area (second stimulation electrode), and the left temporal area (return electrode, see Fig. 1). This electrode configuration was chosen to (i) maximally capture the anterior–posterior direction of the current along the recording electrodes, and (ii) to conceptually replicate previous three-electrode montages[15,18]. During each experiment, the stimulation intensity (0.1 mA peak-to-zero) and frequency (10 Hz) were kept constant for both active stimulation electrodes. At these electrodes, the current density was 0.225 A m$^{-2}$ and the charge density per condition was 6.75 C m$^{-2}$. The return electrode was set to pick up the remaining currents. Low stimulation intensities were chosen to minimize physiological effects possibly confounding the field measurements. Due to Ohm's law electric fields are completely linear with stimulation intensity which means that our results are directly applicable for higher intensities as well[29]. The phase of the stimulation currents was varied systematically starting from 0 degrees phase difference between the two stimulation electrodes and up to 360° in 15° steps. This resulted in 25 phase conditions that were measured in both subjects. Each stimulation condition lasted for 30 s with a ramp up/down time of 5 s.

**Intracranial electric field recordings**. For subject 1, intracranial electric field recordings were performed using a BrainAmp MR plus amplifier (Brain Products) while for subject 2, a Cortech NeurOne Tesla amplifier (Cortech Solution, Wilmington, NC) was used. Ground and reference electrodes were positioned on the scalp over the right temporal area. Both monkeys were anesthetized (capuchin: ketamine 10 mg kg$^{-1}$ IM, atropine 0.045 mg kg$^{-1}$ IM, diazepam 1 mg kg$^{-1}$ IM, isoflurane 2%; macaque: ketamine 12 mg kg$^{-1}$ IM, atropine 0.025 mg kg$^{-1}$ IM, isoflurane 1.25–2% as needed) during the experiments to reduce the stress of the monkeys and keep them still. While anesthesia suppresses neurophysiological activity, it has negligible impact on electric field measurements which are the objective of this study[29].

**Data analysis**. Data were analyzed with MATLAB 2017b (MathWorks) according to previously reported processing steps[29,30]. Data preprocessing included bandpass filtering from 5 to 20 Hz with a fourth order zero-phase, forward-reverse Butterworth filter and downsampling to 1 kHz using the Fieldtrip toolbox[63]. These cutoff frequencies allowed us to inspect the quality of the signal (Supplementary Fig. 3) without introducing any phase distortions due to the use of a two-pass filter and long recording time. Further, we extracted the epochs of interest which correspond to the stimulation conditions (30 s per epoch) from the continuous recordings. As a result, 25 epochs were defined for each subject. Finally, we demeaned the data and visually inspected them for the presence of potential technical artifacts. For the subject 1, we excluded three contacts and interpolated them from neighboring sites due to excessive noise at these contacts. We further excluded the most posterior contact for each electrode because they were outside the gray matter. For the subject 2, two contacts were excluded and interpolated from the neighboring sites. One electrode was entirely removed from the analysis

due to low recording quality. Finally, we rescaled the data (i.e., multiply by 10) to a 1 mA peak-to-zero stimulation current according to reporting convention for neuromodulation studies[12]. Exemplary data are displayed in Supplementary Fig. 3.

After preprocessing, we calculated the amplitude and phase of the recorded TES voltages. The time-series were zero-padded to a length of 32,768 ($2^{15}$) samples. Then, we calculated the Fast Fourier Transformation and extracted the phase angles ($\varphi$) at the stimulation frequency (10 Hz). We unwrapped the phase angles, centered them between $-2/\pi$ and $+2/\pi$, and converted them from radians to degrees. To estimate the maximum phase difference in the brain for a given stimulation condition ($\Delta\varphi$), we subtracted the phase angle from the most posterior recording electrode from the most anterior recording electrode.

In a second analysis step, we calculated the TES-induced electric fields in the brain by computing the numerical gradient of the recorded voltages along the implanted electrodes. Given that the recording electrodes are parallel to the connecting line between two stimulation electrodes, our method accurately captures the dominant anterior-posterior component of the electric field in the brain. Within this component, the field is restricted to two directions. When the directions at every sampling point are identical, we call it a unidirectional electric field, otherwise—bidirectional. Like the TES voltages, the TES electric field is oscillating over time. We estimated the strength of the oscillating electric field as the root mean square amplitude per cycle. This resulted in a more robust estimate compared to the peak amplitude, however, with results qualitatively unaffected by this choice. The phase angles ($\varphi$) of the electric field were calculated in the same way as for the voltage described above.

One intriguing possibility with multi-electrode, multi-phase TACS that we discovered during exploratory data analysis is that the spatial location of maximum electric fields changes with time (i.e., a traveling wave). We thus quantified this traveling wave property as a function of stimulation phase. For this, we first normalized the electric field time-courses for each stimulation condition with respect to their individual maximum. Then, we estimated their pairwise dissimilarity as the mean squared difference across contacts over time and averaged them across electrodes. Dissimilarity close to zero indicates that every recording contact in each electrode detects the highest (or lowest) field magnitudes at the same time point. Thus, the electric field maximum does not move in space over time. High dissimilarity indicates that different contacts detect the electric field maximum at the different time points, i.e., the field's maximum is moving over time (traveling wave).

To estimate the spatial frequency and propagation speed of the TACS-induced "traveling waves", we fitted a linear regression $\varphi_E(x) = r * x + b$, where $r$ is the spatial frequency in deg mm$^{-1}$, $\varphi_E(x)$ is the observed phase of the electric field in degrees along the electrode contacts $x$ (coordinates in mm), and $b$ is the intercept. The wave speed (in mm ms$^{-1}$) $c = f \times 1/r$, where the temporal frequency (in deg ms$^{-1}$) $f = d\varphi_E(x)/dt$.

**Statistical analysis**. To assess the statistical significance of differences in electric field magnitudes between stimulation conditions we implemented a nonparametric one-way ANOVA test (also known as Kruskal–Wallis test). For measurements where the range of values, rather than the absolute values themselves, are meaningful we utilized a one-way ANOVA that compares the absolute deviations from the group median (Brown–Forsythe test). For circular data, such as the phase angles, the absolute deviations from the group mean were tested using a one-way ANOVA for circular data (Watson–Williams multi-sample test). $F$-statistics, degrees of freedom and $p$ values are reported in the results section.

Nonlinear relationships between the measurement results and stimulation conditions were examined using the MATLAB Curve fitting toolbox. We tested the fit of a linear function ($y = a_1 x + b$) and sinusoidal functions ($y = a_0 + a_1\cos(x\omega) + b_1\sin(x\omega)$). The adjusted $R$-squared ($R^2_{adj}$) metrics of goodness-of-fit is reported and complemented by the SSE where appropriate.

**Computer simulations of the electric field**. The main experimental conditions were modeled using the FEM in SimNIBS 2.1[54]. A realistic, FEM model of a monkey head (*Cebus apella*, 13 y.o., male, 4.1 kg), including 6 tissue types: skin and soft tissues ($\sigma = 0.465$ S m$^{-1}$), eyes ($\sigma = 0.5$ S m$^{-1}$), bones ($\sigma = 0.01$ S m$^{-1}$), CSF ($\sigma = 1.654$ S m$^{-1}$), gray matter ($\sigma = 0.275$ S m$^{-1}$), and white matter ($\sigma = 0.126$ S m$^{-1}$) was created. Tissues were segmented from an anatomical MR image using FSL[64], transformed into 3D surfaces with Freesurfer[65], and meshed with Gmsh[66]. Three round electrodes (radius = 1 cm, thickness = 2 mm, gel thickness = 3 mm, gel $\sigma = 1$ S m$^{-1}$, central connector with radius = 0.25 cm) were positioned in the same way in the model as during the real experiments. Every time-step (36 steps per cycle) was modeled separately, and simulations results processed in MATLAB.

**Visualization**. All 2D plots were created in MATLAB (MathWorks) and 3D brain plots—in Gmsh[66]. For each animal, cortical surfaces were reconstructed in the following steps. First, we employed a bias field correction of the T1-weighted images using ANTs (https://stnava.github.io/ANTs) and averaged them across multiple acquisitions. Brain extraction and tissue segmentation was performed using FreeSurfer (https://surfer.nmr.mgh.harvard.edu) and manual corrections using ITK-SNAP (http://itksnap.org). Finally, the white matter and pial surfaces were reconstructed using FreeSurfer. Supplementary Figure 4 was drawn with the Gramm toolbox (https://github.com/piermorel/gramm). All figures are assembled in panels in Inkscape (https://inkscape.org). Note that for 10 Hz TES the length of a single cycle is 100 ms which defines the time window for the corresponding figures and movies.

**Reporting summary**. Further information on research design is available in the Nature Research Reporting Summary linked to this article.

## Data availability
The data that support the findings of this study are available from the corresponding author upon reasonable request.

## Code availability
The custom computer code used in this work is available from the corresponding author upon reasonable request.

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

## Acknowledgements

Research reported in this publication was supported by NIH (R21 MH110217-01, R01 MH111439-01, and P50 MH109429) and the University of Minnesota's MnDRIVE Initiative. We thank Stan Colcombe, Raj Sangoi, and Caixia Hu for MR imaging support and we thank Deborah Ross, Mark Klinger, Kathleen Shannon, and Tammy McGinnis for veterinary assistance.

## Author contributions

A.O. designed the experiments; A.Y.F., G.L. and A.O. collected the data; T.X. processed the MR data; C.E.S. and M.P.M. supervised the data collection; I.A. analyzed the data and prepared the figures; A.O. supervised the data analysis; I.A. and A.O. interpreted the results; I.A. and A.O. wrote the paper; all authors reviewed the paper.

## Additional information

**Competing interests:** A.O. is an inventor on patents and patent applications describing methods and devices for noninvasive brain stimulation. The remaining authors declare no competing interests.

