## [Transparent Peer Review File · Nature Communications]

Reviewers' comments:

Reviewer #1 (Remarks to the Author):

Peer review: NCOMMS-18-18292

"Electric field dynamics in the brain during multi-electrode transcranial electric stimulation"
By Alekseichuk et al.

The manuscript reports on an electrophysiological study investigating the effects of transcranial electric stimulation (tACS) on intracranially recorded local field potentials in two non-human primates. Electrical stimulation is applied through a multi-electrode montage, with three electrodes (two active and one return electrode). The phase between the two active electrodes is shifted systematically over conditions, so that the current flow at the return electrode is dependent on the phase difference between the two active electrodes.

The manuscript is written clear and precise. The experimental design is well-thought. The figures are very clear and of high quality.

Similar montages have been applied recently in human and animal studies, in order to accomplish in- and anti-phase stimulation with the aim to alter coherence between two or more brain areas. The rationale behind these stimulation montages is that connectivity between brain areas is changed with 0° and 180° phase shift between stimulation sites. An assumption which has not yet been tested experimentally with intracranial recordings.

Therefore, the manuscript is extremely helpful for the transcranial stimulation community and provides useful insights into the actual currents in the brain.

Comments:

General

1. The amplitude of the electrical stimulation is quite low (0.1 mA peak-to-zero). Is this low intensity expected to modulate neuronal activity? Would this low amplitude be sufficient to elicit action potentials or subthreshold modulation?
2. A simulation of the current flow would be helpful to evaluate whether the applied tACS can be modeled with current headmodels. What is the expected current flow at the different points in space?
3. In a simulation it would also be interesting to see the direction of dipoles and the change of direction dependent on the phase difference between the active electrodes.
4. It is unclear which brain areas would be the target of such a stimulation. Normally a target is defined and then electrode placement (even with multi-electrode montages) is optimized to target this region or in the case of connectivity modulation to optimize coherence between two regions. The authors could state explicitly whether specific target areas were defined.
5. The finding of a traveling wave is potentially interesting, but the application is still unclear. The authors could speculate a bit more what an adequate application for the traveling wave stimulation would be.

Methods

6. Please provide the current density and the charge density at the electrodes.
7. Which type of electrode was used? Pistim by Neuroelectronics?
8. Page 4: Is it SigmaGel or SignaGel (Parker Labs)?
9. Page 6: Please specify the filter order of the Butterworth filter
10. Why was a narrow bandpass filter (5-20 Hz) used, which can potentially introduce phase distortions? The FFT at 10 Hz in the further analysis extracts the signal only at 10 Hz anyway, so there is no need for a narrow bandpass filter before.

11. Please explain why the electrophysiological signal was multiplied by 10. The current source densities are probably not linearly scaled with the stimulation intensities.

Results

12. Page 10: For the calculation of the effect of condition on the measured phase angles of voltage and electric field a one-way ANOVA is computed. However, the use of circular statistics is necessary if the dependent variable is angular.

13. Supplementary Figure 1: The colors of the line plots (blue, red, and grey) are not well distinguishable (at least in a printed version).

Discussion

14. Page 14: The sentence "This will also apply to four- and electrode montages..." needs some revision. High-density montages for stimulation do not place one electrode over each brain area of interest, but electrode positions are optimized to stimulate the target maximally. See papers by Dmochowski et al. (2011) *J Neural Eng.* and Wagner et al. (2016) *SIAM J Appl. Math.*

Reviewer #2 (Remarks to the Author):

In the manuscript "ELECTRIC FIELD DYNAMICS IN THE BRAIN DURING MULTI-ELECTRODE TRANSCRANIAL ELECTRIC STIMULATION", Alekseichuk et al. show the results of stereotactic EEG measurements during transcranial alternating current stimulation (TACS) performed using 3 electrodes at various phase differences.

The authors show the relationship between electrode phase and the the phase between the electric potentials and electric fields in the cortex. They further show that by tuning the phase difference between the electrodes, "traveling wave" patterns in the cortex appear.

While the paper presents some interesting ideas, I believe that they are not sufficiently well explored. Therefore, I recommend for the manuscript to be rejected and submitted to a more specialized journal after the points bellow have been addressed.

The authors state several times in the paper a "non-linear" relationship between electrode phase differences and the electric field phase. I believe that this is a rather trivial finding for people familiar with volume conductors. The relationship is highly dependent on the location of the electrodes, which is not addressed in the paper.

It seems that a significant proportion of the contacts is located deep in white matter. Please elaborate on the position of the contacts.

The distance between the contacts is of 5mm, which is quite a large distance in the cortex, especially for NHPs. Please discuss how this distance affects the accuracy of the electric field estimates.

The analysis is entirely based on recording electrodes placed along the posterior-anterior axis. And the analysis of the electric field is solely based on this component. However, the TACS return electrode is placed laterally, and therefore I expect that there will be a significant electric field in the left-right direction, especially for phase differences around 0 / 360 degrees. Please discuss how this can affect your conclusions.

The placement of the stimulation and recording electrodes also explains the seemingly contradictory finding that the electric field in the anti-phase (180 degrees) condition is higher than the in-phase (0 / 360 degrees) condition. Even though the total current injected in the later is two times higher than in the former, it is oriented laterally and therefore not captured by the returning electrodes.

The words "anti-directional" and "unidirectional" are used. To the best of my knowledge, these words are not used in the literature and have no immediate obvious meaning, so they should be introduced. I also believe that the use of the term "direction" is misleading, as the recordings are

only able to capture the electric field component along one direction, the term "component" would be more adequate.

The claim by the authors that they have generated a traveling wave pattern is not backed up by the data shown in the paper. All the analysis is done in the normalized electric fields. However it is plausible that the normalized maximum moves while the absolute maximum moves very little or not at all. The authors still claim that "the spatial locations of their electric field maxima are traveling over time" (page 12/13) and have similar statements elsewhere in the paper, which is highly misleading. Please perform the analysis in the "raw" electric fields. Also, it should be clearly stated that only one electric field direction is analyzed.

The authors demonstrate that the traveling wave pattern (of the normalized electric field) happens along the electrodes. However, I expect generating traveling waves along the cortical surface to be a lot harder, as it possesses complex foldings. This is made even more challenging when considering the direction of the electric field relative to the cortex.

Importantly, No physiological effects are demonstrated for the traveling wave pattern.

Because of its exploratory nature, I believe this study could have been performed just as well, if not better, using simulations instead of recordings. This would allow for more freedom to choose electric field components and locations to analyze.

In short: The way the study has been performed lacks generalizability. Only very few sampling points, all along the anterior/posterior axis are analyzed. Only a single component of the electric field is measured, but the authors often refer to their findings as if the electric field has been exhaustively sampled (eg: "We found that the electric field strength is significant larger (2-2.3 times) for the anti-phase than for the in-phase stimulation condition.", page 14) For any of the non-trivial findings of the paper to be confirmed, more montages, more sampling points, and more electric field components need to be analyzed. The analysis of the "traveling wave" is misleading and incomplete, as only the normalized fields are shown. There are no reports of the physiological impact of such novel type of stimulation. The authors present a few good ideas, but only very superficial data, and key elements for a systematic analysis are missing.

We thank both reviewers for their time and efforts. Please find below our point-by-point response (in black) to the reviewer's questions (in blue) below. Direct quotes from the revised manuscript are highlighted in *italic*.

To summarize the changes made in our manuscript, we (i) developed a computational model of multielectrode TES corroborating our experimental findings and extending them to a 3D level of investigation, (ii) conducted an additional experiment with a new configuration of stimulation electrodes, thus generalizing our initial findings, and (iii) performed additional analyses based on the suggestions of both reviewers. All additional results support our original conclusions.

Reviewer #1

We very much appreciate the reviewer's evaluation of the quality and importance of the work. As suggested, we have developed a current flow simulation, which agreed with the experimental results and clarified the methods and statistics. Below we provide a point-to-point response to all comments raised.

1. The amplitude of the electrical stimulation is quite low (0.1 mA peak-to-zero). Is this low intensity expected to modulate neuronal activity? Would this low amplitude be sufficient to elicit action potentials or subthreshold modulation?

This study aimed to address the gap in knowledge regarding the biophysics of stimulation, rather than the physiology, which is explored in a multitude of studies (e.g., from Terzuolo and Bullock, PNAS, 1956; to Vöröslakos et al., Nat. Comm., 2018). We would not expect physiological effects from this low amplitude stimulation and have deliberately chosen it to avoid possible confounds. Thus, the low intensity is fully justified by our primary aim of focusing on the biophysical effects. Due to Ohm's law our findings are directly applicable to higher intensity stimulation aimed at eliciting physiological changes. We have now added to the method section:

“Low stimulation intensities were chosen to minimize physiological effects possibly confounding the field measurements. Due to Ohm's law our results are directly applicable for higher intensities as well.”

In the discussion, we further consider the issue of electric field strength and physiological response:

“... the findings of the present work address important biophysical aspects of multielectrode TES. Previous studies investigated the effect of electric fields on the neurophysiology using

various experimental approaches from in silico⁴⁹ to in vitro^{50,51} and in vivo⁵². Seminal work from Terzolo and Bullock (1956) demonstrated the sensitivity of neurons to weak electric fields ($\approx 1\text{mV/mm}$) strongly depending on the electric field orientation with respect to the somatodendritic axis. More recent studies show the importance of cellular and network level states, and their coherence with the stimulation parameters⁴⁹⁻⁵². However, surprisingly little is known about the electric fields that TES produces in the brain. System level effects of TES are driven by physical factors such as “where” and “when” the generated electric fields interact with neural tissue. While the “where” question has been explored in previous studies⁵⁴⁻⁵⁶, the “when” question specific for alternating current stimulation has been largely missing thus far.

2. A simulation of the current flow would be helpful to evaluate whether the applied tACS can be modeled with current headmodels. What is the expected current flow at the different points in space?

Following the reviewer’s suggestion, we performed a current flow simulation. For this, we developed a realistic NHP model and performed simulations at varying timepoints using SimNIBS 2.1. The electric field was simulated at every time-point for two “standard” conditions (0 deg and 180 deg) and for the traveling wave condition (45 deg). The simulation results agree with the spatial distribution and time-course of the electric field observed during the real experiments. Thus, it is possible to use computational models to describe and further explore phenomena found in our measurements. Please see the new Figure 7 below, and supplementary animations 5-7 that demonstrate the direction and distribution of the electric field on the surface of the grey matter.

3. In a simulation it would also be interesting to see the direction of dipoles and the change of direction dependent on the phase difference between the active electrodes.

We have implemented these suggestions in our model. These results are presented in the supplementary animations 5-7. All simulation results are in agreement with our experimental data and conclusions.

4. It is unclear which brain areas would be the target of such a stimulation. Normally a target is defined and then electrode placement (even with multi-electrode montages) is optimized to target this region or in the case of connectivity modulation to optimize coherence between two regions. The authors could state explicitly whether specific target areas were defined.

Here we positioned the stimulation electrodes in a way that (1) conceptually replicate previously used 3-electrodes montages [Polania et al., Curr. Biol., 2012; Violante et al., eLife, 2017] and (2) maximize the anterior-posterior current flow along the recording electrodes in the brain. These montages aimed to modulate global neocortical connectivity between the prefrontal and

posterior regions, rather than any given sub-area. This point is now included in the method section, chapter “Transcranial Alternating Current Stimulation”:

This electrode configuration was chosen to (i) maximally capture the anterior-posterior direction of the current along the recording electrodes, and (ii) to conceptually replicate previous three-electrode montages^{15,18}.

5. The finding of a traveling wave is potentially interesting, but the application is still unclear. The authors could speculate a bit more what an adequate application for the traveling wave stimulation would be.

We expand the discussion to highlight possible applications:

Such propagation of electric fields resembles the well-known electrophysiological phenomenon of traveling waves in the brain^{44,45}. At the macroscopic level, these traveling waves of electrical activity contribute to the formation of large-scale cell assemblies and neural networks⁴⁶. Traveling waves are observed and functionally relevant in different frequency bands, brain states and anatomical regions, including theta activity in the hippocampus⁴⁷, beta activity in the motor cortex⁴⁸ and gamma activity in the visual cortex⁴⁹. Traveling waves can even occur on a global level spanning the whole neocortex^{50,51}. Recently it was shown that the consistency and power of large-scale traveling waves in the human neocortex are reliably correlated with the performance of an ongoing memory task⁴⁶ and with memory consolidation during sleep⁵². Traveling wave TES thus presents a novel tool to mimic and manipulate such naturally occurring brain activity, obtain causal rather than correlational evidence for its role, and eventually converge into a novel therapeutic approach for cognitive deficits and neuropsychiatric disorders.

6. Please provide the current density and the charge density at the electrodes.

At both active stimulation electrodes, the current density was 0.225 A/m^2 (estimated as the root mean square of the AC amplitude divided by the electrode surface area) and the charge density per condition (i.e., per 30 s) was 6.75 C/m^2 . We have added this information to the methods section:

During each experiment, the stimulation intensity (0.1 mA peak-to-zero) and frequency (10 Hz) were kept constant for both active stimulation electrodes. At these electrodes, the current density was 0.225 A/m^2 and the charge density per condition was 6.75 C/m^2 .

7. Which type of electrode was used? Pistim by Neuroelectronics?

The reviewer is correct, we used Pistim electrodes. We added this information into the Methods.

8. Page 4: Is it SigmaGel or SignaGel (Parker Labs)?

We thank the reviewer for pointing out this typo. Indeed, we used SignaGel (Parker Labs) and have corrected this in the manuscript.

9. Page 6: Please specify the filter order of the Butterworth filter

We used a 4th order two-pass (forward-reverse) Butterworth IIR filter. The information was added to the paper:

Data preprocessing included bandpass filtering from 5 to 20 Hz with a fourth 4th order zero-phase, forward-reverse Butterworth filter...

10. Why was a narrow bandpass filter (5-20 Hz) used, which can potentially introduce phase distortions? The FFT at 10 Hz in the further analysis extracts the signal only at 10 Hz anyway, so there is no need for a narrow bandpass filter before.

The data were bandpass filtered before the FFT to visually inspect data quality (see Sup. Figure 2) without a low-frequency drift and high-frequency line noise. Please notice that we implemented a zero-phase filter by first filtering the data in forward and then in reverse direction (matlab function `filtfilt.m`). Possible phase distortions are minimized using this procedure. As the reviewer correctly noticed, we later focused at the FFT results at 10 Hz, which makes the exact choice of the filter passband less important. Also, given that our actual data epochs were 30 seconds long, a 5-20 Hz filter is sufficiently broad to satisfy the Heisenberg–Gabor limit. Still, we now tested other filters, e.g., 1-40 Hz, with identical results. We have now added to the manuscript:

These cut-off frequencies allowed us to inspect the quality of the signal (Figure S2) without introducing any phase distortions due to the use of a two-pass filter and long recording time.

11. Please explain why the electrophysiological signal was multiplied by 10. The current source densities are probably not linearly scaled with the stimulation intensities.

The signal was multiplied by ten only for visualization purposes for the convenience of the readers, most of whom are used to see the brain stimulation results relative to a stimulation intensity of 1 mA. Please note that the biophysics of the generated electric fields does not depend on the absolute value of stimulation intensity, but only on the relative intensity and phase per electrode (e.g., it is shown in Opitz et al., Sci. Rep., 2016). Intracranial electric fields

are perfectly linear with the applied stimulation intensities (according to Ohm's law). The same is true for current densities as they are also linearly related to the electric field strength.

12. Page 10: For the calculation of the effect of condition on the measured phase angles of voltage and electric field a one-way ANOVA is computed. However, the use of circular statistics is necessary if the dependent variable is angular.

This is a good point. We agree with the reviewer. The dependence of phase angles (in radians) from the stimulation conditions is now evaluated using a one-way ANOVA for circular data (also known as Watson-Williams multi-sample test) as implemented in the Circular Statistics Toolbox for MATLAB. The results are unaffected by this choice. We have updated the paper:

The conditions strongly affect φ_V (One-way ANOVA for circular data for subject 1: $F_{24,700} = 5.23$, $p = 3.03 \times 10^{-14}$; subject 2: $F_{24,525} = 6.55$, $p < 1 \times 10^{-15}$) and φ_E (subject 1: $F_{24,700} = 97.6$, $p < 1 \times 10^{-15}$; subject 2: $F_{24,525} = 6.61$, $p < 1 \times 10^{-15}$) although in different ways.

13. Supplementary Figure 1: The colors of the line plots (blue, red, and grey) are not well distinguishable (at least in a printed version).

We have revised the color scheme to make the colors more distinct.

14. Page 14: The sentence "This will also apply to four- and electrode montages..." needs some revision. High-density montages for stimulation do not place one electrode over each brain area of interest, but electrode positions are optimized to stimulate the target maximally. See papers by Dmochowski et al. (2011) J Neural Eng. and Wagner et al. (2016) SIAM J Appl. Math.

We agree with the reviewer. This point is expanded, and necessary references are provided:

This will also apply to four- or more electrode montages which apply a single stimulation electrode per brain area of interest^{22,23}. However, further research is needed to investigate Laplacian montages where multiple electrodes are positioned over a single brain area of interest for maximal engagement^{38,39}.

Reviewer #2

We appreciate the time and comments of the reviewer. Below we provide our point-by-point response which will hopefully resolve the reviewer's concerns.

1) The authors state several times in the paper a “non-linear” relationship between electrode phase differences and the electric field phase. I believe that this is a rather trivial finding for people familiar with volume conductors.

We can only disagree with this comment. Actually, our findings are counterintuitive - volume conduction should **reduce**, rather than **introduce** the phase nonlinearities we observe. For frequencies up to 1000 Hz, volume conduction is assumed to be quasi-static, because (i) capacitive components of tissue impedance, (ii) inductive effects, and (iii) electromagnetic propagation effects are negligibly small (Plonsey and Heppner, Considerations on Quasi-Stationarity in Electro-physiological Systems, Bull. math. Biophys., 29, 1967). Our findings are also nontrivial from the perspective of prior research. Review of the existing literature and reveals no previous work describing a similar finding to ours, and no citation to such work is provided by the reviewer.

The major contribution of this paper is the demonstration that through the combination of multiple sources with different phases, even a linear volume conductor can show non-linear behavior which we have demonstrated here. Using a novel combination of phase-shifted input currents, we create a novel stimulation condition previously not presented in the literature. Thus, our study opens up possibilities for TES previously unavailable to the community, which, we think, are highly interesting and certainly not trivial.

We have now added to the manuscript:

Both for the purpose of TES modeling and EEG source analysis, the head is assumed as a quasi-static volume conductor, because in the frequency range of interest (< 1000 Hz) capacitive components of tissue impedance, inductive effects, and electromagnetic propagation effects are negligibly small⁵⁴. However, through the combination of multiple sources with different phases even a linear volume conductor can show non-linear behavior, as we have demonstrated here. 2) The relationship is highly dependent on the location of the electrodes, which is not addressed in the paper.

Of course, this is true, but again, not in a trivial sense. To address this point, we have now added a new dataset for subject #2 where the stimulation electrodes are rotated: one active over the left temple, second active on the forehead, and the return over the occipital lobe. In other words, we turned the electrode-montage and corresponding electric field perpendicular relative to what was used in the paper before. We demonstrate that the traveling wave effect remains present, although the best phase shift of stimulation currents for its generation is

different. Thus, the “traveling wave” stimulation paradigm is generalizable. Also, please note the normalized and “raw” data in the figure.

We have added this figure to the supplementary materials, and necessary references are made in Results. We have added to the manuscript:

To establish the generalizability of the “traveling wave” stimulation, we conducted a control experiment. In subject 2, the stimulation electrode-montage was rotated: one active electrode was located on the left temple, another – on the forehead, and the return electrode – on the back of the head (Fig. S8A). Thus, the generated electric field was mostly perpendicular to the one in the main experiment. However, the phenomenon of “traveling wave” remained present (Fig. S8B, C, D), although with the most optimal phase shift of stimulation currents for its generation also rotated and established at 165° and 195°. Considering future applications of traveling wave TACS, adequate modeling will be necessary to establish the most optimal parameters for a given configuration of the stimulation electrodes.

3) It seems that a significant proportion of the contacts is located deep in white matter. Please elaborate on the position of the contacts.

We have now added a table with the locations of all recording contacts to the supplementary material.

Subject	Electrode	Anatomical locations (from more posterior to more anterior structures)
1	1	Area V1, Area V2, Area V3, Area V4, Ventral intraparietal area, Corpus callosum, Caudate, Brodmann Area 24, Brodmann Area 32, Brodmann Area 25
	2	Area V1, Area V2, Area V3, Area V4, Ventral intraparietal area, Medial superior temporal area, Pulvinar, Frontal eye field
	3	Area V1, Area V2, Area V1 peripheral, Area V2 peripheral, Hippocampus, Parahippocampal area
2	1	Area V1, Areas V2/V4, Area MT, Ventral intraparietal area, White matter, Pulvinar/Caudate, Caudate/Corpus callosum, Caudate, Brodmann Area 25
	2	Area V1, Area V2, Area V3, Area V1 peripheral, Area V2 peripheral, Hippocampus, Area V4, Parahippocampal areas

4) The distance between the contacts is of 5mm, which is quite a large distance in the cortex, especially for NHPs. Please discuss how this distance affects the accuracy of the electric field estimates.

We have chosen these recording electrodes because we wanted to measure synchronization across the whole brain (e.g. frontal-parietal). For this a 5mm spacing is adequate as used in previous studies (Opitz et al., Sci. Rep., 2016; Opitz et al., PNAS 2017, Huang et al., eLife, 2017). Our experimental findings (please see Figures 2-6) demonstrate continuous gradients of voltage and phase differences in the brain during TES in both NHPs. Thus 9 to 11 sampling points (i.e., contacts) per electrode are sufficient to describe the electric field across a larger brain scale, which is the focus of our study.

5) The analysis is entirely based on recording electrodes placed along the posterior-anterior axis. And the analysis of the electric field is solely based on this component. However, the TACS return electrode is placed laterally, and therefore I expect that there will be a significant electric field in the left-right direction, especially for phase differences around 0 / 360 degrees. Please discuss how this can affect your conclusions.

We agree with the reviewer and thus we have expanded our analysis addressing this issue. For this, we developed a simulation of our experimental conditions (shown in Figure 7 and animations 5-7) using SimNIBS 2.1. These models cover all possible directions of the electric field and are in full agreement with the experimental data regarding the space- and time-courses of the electric fields. The electric field was simulated at every time-point for the two “standard” conditions (0 deg and 180 deg) and for the traveling wave condition (45 deg). The results confirm the ability to generate the electric field on the cortical surface. Please see new

Figure 7 below, and supplementary animation 5-7 that demonstrate the direction and distribution of the electric field on the surface.

Further, the electric field mostly affects neurons that are aligned with the electric field and ignores neurons in the perpendicular plane (* Rahman, A., Reato, D., Arlotti, M., Gasca, F., Datta, A., Parra, L.C. and Bikson, M. (2013), Cellular effects of acute direct current stimulation: somatic and synaptic terminal effects. *The Journal of Physiology*, 591: 2563-2578). Hence, for every single neuron only one axis of electric field matters, and electric fields with different directions will affect a different subset of neurons in the brain. Therefore, while we looked only at the anterior-posterior axis experimentally, the changes that we observed are meaningful and comprehensive for all pyramidal neurons in this plane, because they will not be sensitive to other directions of the electric field. We also considered this point in the second paragraph of the Discussion:

Here, we should note that the measured electric fields are restricted to their anterior-posterior component along the implanted electrodes. It is to be expected that in the 0° conditions medial-lateral components will be larger than for the 180° condition. This does not diminish the validity of our findings as the direction of the electric field with respect to the neural elements is most important for TES physiological effects^{40,41}. The electric fields with different directions will affect a different subset of neurons in the brain. Thus, differences in one electric field component across stimulation conditions are important to document and control for. Here, our measurements can inform future experimental work so that it can better control for these effects. Moreover, we performed electric field modeling of our main experimental conditions. The models are descriptive for all directions of the electric field in the brain and are in agreement with our experimental data. This further strengthens the generalizability of our findings.

6) The placement of the stimulation and recording electrodes also explains the seemingly contradictory finding that the electric field in the anti-phase (180 degrees) condition is higher than the in-phase (0 / 360 degrees) condition. Even though the total current injected in the later is two times higher than in the former, it is oriented laterally and therefore not captured by the returning electrodes.

We agree with the reviewer that the recording electrode placement is one of the reasons, which we had stated in the discussion, starting with the phrase: *“Here, we should note that the measured electric fields are restricted to their anterior-posterior component [...]”* (please see answer 5 above).

Further, in the 180° stimulation condition, the current is passing between two electrodes on opposite sides of the head far away from each other. But in the 0° condition, despite the twice larger current, it passes almost twice shorter distances from anterior and posterior electrodes to the middle temporal electrode. As a result, more of it is shunted through the skin. Thus, we see no contradiction.

Please also notice that we have added a realistic computational model to the revised manuscript that captures this effect – both conditions are characterized by the maximum E-field in the grey matter of approx. 2.4 mV/mm for an input of 1 mA per active electrode (see Figure above).

We have expanded on this in the manuscript:

Interestingly, despite a twice stronger current in the 0° than in the 180° stimulation condition, all models indicate the same maximum electric field in the grey matter of 2.4 mV/mm (per 1 mA on

both active electrodes). This is likely due to the higher percentage of shunted current in the otherwise stronger 0° condition, as the involved electrodes on the scalp are closer to each other.

7) The words “anti-directional” and “unidirectional” are used. To the best of my knowledge, these words are not used in the literature and have no immediate obvious meaning, so they should be introduced. I also believe that the use of the term “direction” is misleading, as the recordings are only able to capture the electric field component along one direction, the term “component” would be more adequate.

While indeed the measured electric fields are components (anterior-posterior), they can have different directions. One could imagine this as + or – direction on the x-axis. Thus, we think using the word direction is adequate in this context. We have made this now clearer in the manuscript adding the following definitions while also emphasizing that this is of the anterior-posterior component:

Given that the recording electrodes are parallel to the connecting line between two stimulation electrodes, our method accurately captures the dominant anterior-posterior component of the electric field in the brain. Within this component, the field is restricted to two directions. When the directions at every sampling point are identical, we call it a unidirectional electric field, otherwise – anti-directional.

8) The claim by the authors that they have generated a traveling wave pattern is not backed up by the data shown in the paper. All the analysis is done in the normalized electric fields. However it is plausible that the normalized maximum moves while the absolute maximum moves very little or not at all. The authors still claim that “the spatial locations of their electric field maxima are traveling over time” (page 12/13) and have similar statements elsewhere in the paper, which is highly misleading. Please perform the analysis in the “raw” electric fields.

First, we would like to stress that normalization or rescaling to the same max/min of the time-series is a standard and routine practice in digital signal processing. Among other things, it is a prerequisite for our numerical estimation of a similarity/dissimilarity index (i.e., average mean square deviation), which should not be done otherwise.

In any case, we have now performed the analysis on the “raw” electric fields in the same way as in Figure 6 (shown below, also included as in the paper as Sup. Figure 7.) Please note that that the maximum per individual time-point (each column on the figure) is always at the contact #11

for $0^\circ / 180^\circ$ condition but can be at the contact #2 or #11 for $45^\circ / 315^\circ$ condition. Thus, the “raw” data are in full agreement with the previous conclusion of a changing maximum over time.

9) Also, it should be clearly stated that only one electric field direction is analyzed.

We have stated this clearly multiple times in the paper and have further expanded on this with our added computational model.

(Methods) *“Given that the recording electrodes are parallel to the connecting line between two stimulation electrodes, our method accurately captures the dominant anterior-posterior component of the electric field in the brain.”*

(Discussion) *“Here, we should note that the measured electric fields are restricted to their anterior-posterior component along the implanted electrodes. It is to be expected that in the 0° conditions medial-lateral components will be larger than for the 180° condition. This does not diminish the validity of our findings as the direction of the electric field with respect to neural elements is most important for TES physiological effects^{40,41}. The electric fields with different directions will affect a different subset of neurons in the brain. Thus, differences in one electric field component across stimulation conditions are important to document and control for. Here, our measurements can inform future experimental work so that it can better control for these effects.”*

10) The authors demonstrate that the traveling wave pattern (of the normalized electric field) happens along the electrodes. However, I expect generating traveling waves along the cortical surface to be a lot harder, as it possesses complex foldings. This is made even more challenging when considering the direction of the electric field relative to the cortex.

We understand the reviewer’s concerns and, in order to resolve them, provide new computer simulations of our experiments in SimNIBS. Please see answer 5 above for details, and Figure

7 and supplementary animations 5-7. Our results agree with the experimental data and confirm the ability to generate the electric field on a folded cortical surface.

11) Importantly, No physiological effects are demonstrated for the traveling wave pattern.

We would like to point out that the aim of this study was not related to the physiological effects of TES. Here we addressed the questions of “where” and “when” the electric fields affect the brain, while physiological experiments can answer the question “how” the brain is affected. We are convinced that all these questions are important and necessary; however, they require different approaches and do not diminish one another. Thus, we write in the discussion:

In summary, the findings of the present work address important biophysical aspects of multielectrode TES. Previous studies investigated the effect of electric fields on the neurophysiology using various experimental approaches from in silico⁵⁴ to in vitro^{55,56} and in vivo⁵⁷. Seminal work from Terzolo and Bullock (1956) demonstrated the sensitivity of neurons to weak electric fields ($\approx 1\text{mV/mm}$) strongly depending on the electric field orientation with respect to the somatodendritic axis. More recent studies show the importance of cellular and network level states, and their coherence with the stimulation parameters^{54–57}. However, surprisingly little is known of the present electric fields during TES in the brain. System level effects of TES are driven by physical factors such as “where” and “when” the generated electric fields interact with neural tissue. While the “where” question has been explored in previous studies^{59–61}, the “when” question specific for alternating current stimulation was largely missing thus far. Here, we examined the spatiotemporal properties of multi-electrode TACS with direct, stereotactic recordings in the non-human primate brain.

12) Because of it’s exploratory nature, I believe this study could have been performed just as well, if not better, using simulations instead of recordings. This would allow for more freedom to choose electric field components and locations to analyze.

We respectfully disagree. Because a “traveling wave” electric field effect was never suggested before, simulation approaches were never validated to demonstrate such effect. In addition, we have now expanded our study and include a computational model describing and extending our experimental results (see also answer 5 above).

13) In short: The way the study has been performed lacks generalizability. Only very few sampling points, all along the anterior/posterior axis are analyzed. Only a single component of the electric field is measured, but the authors often refer to their findings as if the electric field has been exhaustively sampled (eg: “We found that the electric field strength is significant larger

(2-2.3 times) for the anti-phase than for the in-phase stimulation condition.”, page 14) For any of the non-trivial findings of the paper to be confirmed, more montages, more sampling points, and more electric field components need to be analyzed. The analysis of the “traveling wave” is misleading and incomplete, as only the normalized fields are shown. There are no reports of the physiological impact of such novel type of stimulation. The authors present a few good ideas, but only very superficial data, and key elements for a systematic analysis are missing.

In the revised version we have addressed all concerns raised by the reviewer:

- (1) An additional data-set with new electrode-configurations is added. It confirms our findings and strengthens their generalizability;
- (2) New computer simulations examining all possible directions and components of the electric field were performed, fully supporting our results and conclusion;
- (3) An analysis of the “raw”, non-normalized data is performed and provided. Its results are in full agreement with the previous findings.
- (4) There clearly are many reports of traveling wave phenomena in the brain, both in support of normal brain operations and as manifestations of pathological conditions such as epilepsy. Ours is the first paper to suggest that we can create TES conditions that will mimic and manipulate these critical neurobiological events.

Reviewers' comments:

Reviewer #1 (Remarks to the Author):

The revision of the manuscript has improved the quality of the manuscript. The purpose of the study, to investigate the biophysical effects of the stimulation and not the neurophysiological effects, is much clearer now. In this respect the study is different from the many studies investigating the physiological effects. Nevertheless, the questions addressed are important for all researchers who are applying transcranial electric stimulation and should be published.

Reviewer #2 (Remarks to the Author):

In the revised version of the manuscript "ELECTRIC FIELD DYNAMICS IN THE BRAIN DURING MULTI-ELECTRODE TRANSCRANIAL ELECTRIC STIMULATION", Alekseichuk et al. elaborated on the methods used and clarified a few points. However, there are still key points missing in the analysis that hinder the understanding of the data. Here, I will present three major points that are still mostly ignored throughout the paper, as well as some minor points to be considered. Therefore, I still feel that the revised version is not suitable for publication in a high-impact journal. In its current form, the paper is still a mostly phenomenological description of the effect, without any attempt to properly analyze its behavior and to confirm that the recorded phase effects fit the theoretically expected pattern. The shown effect of the phases interacting non-linearly is also not an unexpected finding, as it simply follows from basic math.

1) Non-linear phase relationships are the expected behavior, and trivial to see from some basic math: there seems to be a misconception expressed in several points throughout the paper "Line 77: We found a non-linear relationship ..." , "Line 407: Importantly, the electric field magnitude between stimulation conditions is changing in a non-linear manner", and the worst "Line 463: However, through the combination of multiple sources with different phases even a linear volume conductor can show non-linear..." as well as in the response for my first issue. A linear conductor does not have a linear response for all inputs (e.g: the conductivities). In fact, as the authors point out "Line 199: Due to Ohm's law electric fields are completely linear with stimulation intensity which means that our results are directly applicable for higher intensities as well.". That is the only thing that is linear in "linear" conductors. In fact, even the term "linear conductor" is not standard in the literature and should be replaced by "Ohmic conductor". While an experimental validation is always welcome, it is clear from the start that if you have a linear conductor and 2 electric field sources at phases p_1 and p_2 , producing electric fields $E_1(x)$ and $E_2(x)$, we have that

$$E_3(x) \cdot \sin(t+p_3(x)) = E_1(x) \cdot \sin(t+p_1) + E_2(x) \cdot \sin(t+p_2) \quad (R1)$$

It comes from basic trigonometry that $p_3(x)$ is NOT equal to $E_1(x) \cdot p_1 + E_2(x) \cdot p_2$ or any linear combination of p_1 and p_2 . So I keep my point that it should be immediately clear that the relationship between the phases is not linear.

So I suggest a few changes in that regard:

1.1) Change all the references of "linear conductor" to "ohmic conductor". Remove misrepresentations of what the meaning of the linearity is in this context, such as seen in the lines cited above. Do not refer to the effect of the phases interacting non-linearly as an unexpected finding, as all you need is basic math (Eq. R1) to show it is not linear.

1.2) Add some basic mathematical treatment of the electric fields. Phasor algebra, a tool widely used in electrical engineering for many decades, explains many, if not all, findings in the paper.

Namely, phasor sum formulas allows us to calculate the phase and amplitude of the sum of 2 signals of different amplitudes and phases, but the same frequency (Eq. R1). In the context of the paper, the signals would be the potentials "V" or the electric field components "E" caused by each stimulator. As those interact linearly, phasor algebra will tell us what is the output phase and amplitude. You can re-write equation (R1) in terms of the fields obtained with a 0 and 180 degrees phase differences, and use the 0 and 180 degrees data to predict the phase shifts observed at the other phase differences.

2) **Add a basic mathematical analysis to the phenomenological description of the phase relationships:** What you have described as a "traveling wave" is that the output phase "p3" (Eq. R1) increases somewhat linearly along the contact "x" coordinate.

$$p3(x) = r \cdot x$$

Here "r" is the spatial frequency of the traveling wave.

2.1) Please present a proper mathematical treatment of what would be the traveling wave in the studied context.

2.1) Please plot the phase of the fields at each contact as a function of the contact position along the electrode axis, for the various phase differences.

2.2) With the plots above you can do a linear regression to figure out the spatial frequency and wavelength of the traveling wave. And with some additional computations, you can also calculate the speed of the wave. Please compare those to wavelength and speed in naturally occurring cortical waves. If those do not match well, can you really say that it will interact with the naturally occurring cortical waves?

2.3) With all the above, the "dissimilarity index" becomes a very superficial and unclear form to quantify the traveling waves. Substituting it by wavelength or spatial frequency will provide much clearer results.

2.4) With the dissimilarity index out, you want also drop usage of the normalized electric field. It masks the actual data you present by scaling each contact individually, misleading the reader to think that you have a clear sinusoidal wave along the electrodes. When in reality the amplitude of the wave is varying drastically, and the wavelength will probably also vary significantly in space. Please remove all the normalized data from the main paper and have the non-normalized data instead.

3) **The treatment of the simulations by the authors is very superficial. It is not at all shown that the simulations are "in full agreement with the experimental data", as claimed in line 416.**

3.2) Not only because the electric field changes location in time you will have a traveling wave. In addition to that, the movement should be somewhat orderly (p3(x) should be somewhat continuous, not jumping randomly). This was superficially shown for the electrode recordings with the dissimilarity index, even though more analysis is needed (see point 2), but not at all shown for the simulations. A way to do it would be to plot the phase and amplitude at each cortical position. Those can be easily calculated using the phasor sum formulas.

3.1) Simulations show the **norm** of the electric field over time. However, as the authors point out in Line 411 and Line 471, the direction of the electric field is important. So the plots should also probably feature a specific electric field direction, such as the normal direction.

3.2) Working with the normal has other disadvantages. Looking to Eq. R1, it is not clear to me if

the norm of the electric fields will have a well-defined phase. In fact, it is easy to show that in some circumstances the norm of the field can be constant.

3.2) As the $\langle b \rangle$ normal component $\langle b \rangle$ is highly dependent on the local cortical shape, I expect that for the normal component the “traveling wave” will not be present. That is, $p_3(x)$ in Eq. R1 will suffer large discontinuities throughout the cortical surface. Especially when considering humans. Please show some data and discuss the results.

Other points:

O.1) Line 26: Which studies need to be re-evaluated? As far as I know, it is only the small sub-set tACS studies applies “anti-phasic” protocols. And even then, those are limited to 0 and 180 degrees phase differences, as pointed out by the authors in Line 52. The study also shows that there is no traveling wave for 0 degrees and 180 degrees (Line 283). So there will be no studies to be re-evaluated, because this kind of stimulation has never been used. Please remove this sentence from the abstract.

O.2) Line 65: Please be more clear as to what you mean by “translate linearly into phase differences”. As far as I’m aware, it is assumed that in anti-phasic stimulation the phase is shifted by 180 degrees in the regions underneath the electrodes.

O.3) Line 175: I think that the term “anti-directional” is inappropriate. Please change it to “bidirectional” or something else with a more clear meaning. .

O.4) In the paragraph started in line 269, there are several mentions to the voltage and electric field “data”, and a mention to a non-linear increase and decrease. To which data more precisely are the authors referring to? From the context I infer it would be the rms or the maximum electric field for each phase and in each electrode, but it is not clear. The same is valid elsewhere in the text, such as in the legend of several plots.

O.5) I could not find the new animations to download. Were they uploaded?

O.6) It should be made clear that this difference between the in- and anti-phase conditions is highly dependent on the electrode set-up.

O.7) Several points in the paper, it is said that the maximum moves. By how much does it move (in mm)? From the raw data in supplementary 7 and 8 I would think it moves little to nothing, and often simply jump between far away contacts. This does not mean you do not have a wave, it just means that the amplitude of the wave changes drastically. But all mentions to the “maxima moving over time” should be rephrased to a less misleading sentence.

O.8) Line 409: The authors misrepresent would be the goal of the “in-phase” and “anti-phase” simulation. In the “in-phase” conditions, fields are supposed to be simultaneously excitatory on inhibitory. As the electrodes are positioned in opposing ends of the head and the important field component is the normal component, the electric fields should be “anti-directional”. One more reason why the term “anti-directional” is unfortunate.

O.9) Line 448: No evidence is presented that the proposed stimulation protocol actually does mimic the physiological traveling wave. It might be that, in a more realistic setting, the kind of activity caused by the proposed stimulation protocol is different from the cortical traveling waves.

We want to thank the reviewers for their constructive comments. In response, we have further expanded the analysis of our data in combination with additional computational modeling as requested by reviewer 2. This embeds our experimental data in a larger analytical framework which will help other researchers build on our findings. As requested, we focus our response to reviewer 2 on the technical comments made by the reviewer. All changes in the manuscript are highlighted with yellow marker.

Below is our point-by-point response (in black) to the remaining questions (in blue). Direct quotes from the revised manuscript are highlighted in *italic*.

Reviewer #1 (Remarks to the Author):

The revision of the manuscript has improved the quality of the manuscript. The purpose of the study, to investigate the biophysical effects of the stimulation and not the neurophysiological effects, is much clearer now. In this respect the study is different from the many studies investigating the physiological effects. Nevertheless, the questions addressed are important for all researchers who are applying transcranial electric stimulation and should be published.

We very much appreciate the reviewer's support of our work.

Reviewer #2 (Remarks to the Author):

In the revised version of the manuscript "ELECTRIC FIELD DYNAMICS IN THE BRAIN DURING MULTI-ELECTRODE TRANSCRANIAL ELECTRIC STIMULATION", Alekseichuk et al. elaborated on the methods used and clarified a few points. However, there are still key points missing in the analysis that hinder the understanding of the data. Here, I will present three major points that are still mostly ignored throughout the paper, as well as some minor points to be considered. Therefore, I still feel that the revised version is not suitable for publication in a high-impact journal. **In its current form, the paper is still a mostly phenomenological description of the effect, without any attempt to properly analyze its behavior and to confirm that the recorded phase effects fit to the theoretically expected pattern. The shown effect of the phases interacting non-linearly is also not an unexpected finding, as it simply follows from basic math.**

1) **Non-linear phase relationships are the expected behavior, and trivial to see from some basic math:** there seems to be a misconception expressed in several points throughout the

paper “Line 77: We found a non-linear relationship ...” , “Line 407: Importantly, the electric field magnitude between stimulation conditions is changing in a non-linear manner”, and the worst “Line 463: However, through the combination of multiple sources with different phases even a linear volume conductor can show non-linear...” as well as in the response for my first issue. A linear conductor **does not** have a linear response for all inputs (e.g: the conductivities). In fact, as the authors point out “Line 199: Due to Ohm’s law electric fields are completely linear with stimulation intensity which means that our results are directly applicable for higher intensities as well.”. That is the only thing that is linear in “linear” conductors. In fact, even the the term “linear conductor” is **not standard** in the literature and should be replaced by “Ohmic conductor”. While an experimental validation is always welcome, it is clear for from the start that if you have a linear conductor and 2 electric field sources at phases p_1 and p_2 , producing electric fields $E_1(x)$ and $E_2(x)$, we have that $E_3(x) \cdot \sin(t+p_3(x)) = E_1(x) \cdot \sin(t+p_1) + E_2(x) \cdot \sin(t+p_2)$ (R1)

It comes from basic trigonometry that $p_3(x)$ is NOT equal to $E_1(x) \cdot p_1 + E_2(x) \cdot p_2$ or any linear combination of p_1 and p_2 . So I keep my point that it should be immediately clear that the relationship between the phases is not linear. So I suggest a few changes in that regard:

1.1) Change all the references of “**linear conductor**” to “**ohmic conductor**”. Remove misrepresentations of what the meaning of the linearity is in this context, such as seen in the lines cited above. **Do not refer to the effect of the phases interacting non-linearly as an unexpected finding, as all you need is basic math (Eq. R1) to show it is not linear.**

We acknowledge the notion that the brain stimulation field uses a nomenclature different from electrical engineering. We think that we have reconciled this in the revised manuscript. Following the reviewer’s suggestion, we have adopted the term “ohmic conductor” throughout the manuscript when putting the results in a larger framework. The use of the word “nonlinear” is simply a description of the trend in the measured data. Given a large tACS literature (Polania et al., Current Biology, 2012, Nature Communications, 2015; Violante et al., eLife, 2017; Tseng et al., Scientific Reports, 2018; etc.) which has employed these methods, a careful characterization of these effects is crucial for the development of a mechanistic understanding of multi-electrode tACS. We have further expanded the theoretical section outlining the underlying equations and including an additional computational model (see answers below). In addition, we have reformulated the paragraphs mentioned by the reviewer:

(page 3, previously line 77) *“We show in vivo a non-linear relationship between the phase difference of transcranially applied currents and the phases and magnitude of measured intracranial electric fields.”*

(page 15, previously line 463) *“Both for the purpose of TES modeling and EEG source analysis, the head is assumed as a quasi-static volume conductor, because in the frequency range of interest (< 1000 Hz) capacitive components of tissue impedance, inductive effects, and electromagnetic propagation effects are negligibly small⁵¹. However, through the combination of multiple sources with different phases, it is possible to control the local phase and create phase gradients across remote brain regions.”*

1.2) Add some basic mathematical treatment of the electric fields. Phasor algebra, a tool widely used in electrical engineering for many decades, explains many, if not all, findings in the paper. Namely, phasor sum formulas allows us to calculate the phase and amplitude of the sum of 2 signals of different amplitudes and phases, but the same frequency (Eq. R1). In the context of the paper, the signals would be the potentials “V” or the electric field components “E” caused by each stimulator. As those interact linearly, phasor algebra will tell us what is the output phase and amplitude. You can re-write equation (R1) in terms of the fields obtained with a 0 and 180 degrees phase differences, and use the 0 and 180 degrees data to predict the phase shifts observed at the other phase differences.

As suggested, we have expanded the mathematical analysis of the expected electric fields arising from phase-shifted multi-electrode TACS using phasor algebra. These results are presented in the Supplementary Materials as “Supplementary Discussion”:

For any given phases φ_1 and φ_2 of the input currents applied through the two stimulation electrodes with a common reference, we can determine the resulting electric field in the brain based on the superposition principle. For this, we transform the electric fields \vec{E}_1 and \vec{E}_2 resulting from stimulation electrode 1 with phase φ_1 and stimulation electrode 2 and phase φ_2 (with a common reference electrode) into their phasor form P_1 and P_2 . Below we derive the equations for the x-component of the electric field (y and z components can be derived in the same way):

$$P_1^x = E_1^x \cos \varphi_1 + j(E_1^x \sin \varphi_1)$$

$$P_2^x = E_2^x \cos \varphi_2 + j(E_2^x \sin \varphi_2)$$

To calculate the resulting electric field arising from three-electrode stimulation, we can exploit the superposition principle:

$$P_3^x = (E_1^x \cos \varphi_1 + E_2^x \cos \varphi_2) + j(E_1^x \sin \varphi_1 + E_2^x \sin \varphi_2)$$

From the resulting phasor P_3 we can estimate the phase φ_E^x and amplitude $|E^x|$ of the resulting electric field as:

$$\varphi_E^x = \tan^{-1} \frac{(E_1^x \sin \varphi_1 + E_2^x \sin \varphi_2)}{(E_1^x \cos \varphi_1 + E_2^x \cos \varphi_2)}$$

$$|E^x| = \sqrt{(E_1^x \cos \varphi_1 + E_2^x \cos \varphi_2)^2 + (E_1^x \sin \varphi_1 + E_2^x \sin \varphi_2)^2}$$

Thus, phasor analysis provides a way for the precise characterization of phase relationships of the electric field during multi-electrode TACS with any arbitrary configuration of input currents. This approach enables future efforts for optimizing TACS electrode configurations to create specific phase differences across brain regions.

2) Add a basic mathematical analysis to the phenomenological description of the phase relationships: What you have described as a “traveling wave” is that the output phase “ p_3 ” (Eq. R1) increases somewhat linearly along the contact “ x ” coordinate.

$p_3(x) = r \cdot x$. Here “ r ” is the spatial frequency of the traveling wave.

2.1) Please present a proper mathematical treatment of what would be the traveling wave in the studied context.

Following the reviewer’s suggestion, we have estimated the spatial frequency of the traveling wave. For that, we performed a linear regression $\varphi_3(x) = r \cdot x + b$, where r is the spatial frequency in deg/mm, $\varphi_3(x)$ is the observed phase in degrees along the electrode contacts x (in mm), and b is the intercept. Please find below the results (now added to the result section of the manuscript):

(page 11) *Mathematically, “traveling wave” can be described with a spatial phase gradient. Using linear regression of the electric field phases φ_E along the recording electrode contacts we can estimate the spatial frequency r and propagation speed c of the traveling wave³³. We found the median spatial frequency r for subject 1 as 1.42 deg/mm (range: 0.06-4.8 deg/mm), 1.28 deg/mm (0.08-5.88 deg/mm), and 1.06 deg/mm (0.01-5.8 deg/mm) along electrodes 1, 2, and 3 across phase conditions. The median spatial frequency r for subject 2 is 1.5 deg/mm (0-4.25 deg/mm) along electrode 1 and 1.49 deg/mm (0.07-5.57 deg/mm) along electrode 2. The spatial frequency is almost zero for stimulation with $180^\circ \pm 60^\circ$ current phase shift. We interpret $r \geq 1$ deg/mm as a sufficient gradient to detect a traveling wave^{33,34}. This analysis agrees with the*

dissimilarity index (Figure 6a, b). The median propagation speed c of the traveling waves is 1.64 mm/ms (range: 0.75-3.99 mm/ms), 1.63 mm/ms (0.61-4.19 mm/ms), and 2 mm/ms (0.62-5.74 mm/ms) along electrodes 1, 2, and 3 in subject 1; and 1.57 mm/ms (range: 0.85-3.82 mm/ms) and 1.47 mm/ms (0.65-4.1 mm/ms) along electrodes 1 and 2 in subject 2 [...]

Given that the traveling wave, by definition, should exhibit a phase gradient (Muller et al., Nature Reviews Neuroscience, 2018), we consider any condition that generates a spatial frequency of more than 1 deg/mm as satisfying the traveling wave criteria (in range with physiological observations, e.g., Zhang et al., Neuron, 2018). Our results agree with our previous analyses which indicated that stimulation at a 180° phase shift and adjacent conditions lead to almost zero phase differences between the most anterior and most posterior points of observations (Figure 4b) and almost zero dissimilarity index that we used to identify the traveling wave (Figure 6a-b).

Figure 4. Voltage and electric field phases in the brain during TES for a given stimulation condition. **b** The difference between the electric field phase differences at the most anterior and most posterior contact ($\Delta\phi_E$) per each electrode. The left figure corresponds to subject 1, and the right figure to subject 2.

Figure 6. Traveling wave stimulation. Electric field time-courses across different recording contacts (= anatomical locations) are identical for 0°/360° and 180° stimulation conditions but demonstrate traveling wave properties for intermediate stimulation conditions e.g. 45°. **a-b** Dissimilarity (mean square difference) between the electric field time-courses at different contacts.

We have added the following description to the method section:

(page 18) To estimate the spatial frequency and propagation speed of the TACS-induced “traveling wave”, we fitted a linear regression $\phi_E(x) = r * x + b$, where r is the spatial frequency in deg/mm, $\phi_E(x)$ is the observed phase of the electric field in degrees along the electrode

contacts x (coordinates in mm), and b the intercept. The wave speed (in mm/ms) is calculate as $c = f \times 1/r$, with the temporal frequency (in deg/ms) $f = d\varphi_E(x) / dt$.

2.1) Please plot the phase of the fields at each contact as a function of the contact position along the electrode axis, for the various phase differences.

As requested, we have added this plot as Supplementary Figure 6. Note that the distance between the adjacent electrode contacts is constant and equal to 5 mm. As it can be seen, stimulation with phase shifted currents (e.g. 45°) induces a smooth and gradual change of the electric field phase along the electrode contacts.

Supplementary Figure 6. Local electric field phase in the brain during TES for a given stimulation condition in subject 1 (a) and 2 (b). The first contact for each electrode corresponds to the most posterior location, and the last contact – to the most anterior location, with 5 mm spacing between adjacent contacts. This figure complements Figure 4 in the main paper.

2.2) With the plots above you can do a linear regression to figure out the spatial frequency and wavelength of the traveling wave. And with some additional computations, you can also calculate the speed of the wave. Please compare those to wavelength and speed in naturally occurring cortical waves. If those do not match well, can you really say that it will interact with the naturally occurring cortical waves?

As outlined in question 2.1 above, we performed the linear regression and estimated the spatial frequency. Then, we estimated the propagation speed for the stimulation conditions in which the traveling wave are present (i.e., where the spatial frequency ≥ 1 deg/mm). Below are the estimates which we have also added to the result section of the manuscript:

(page 11) The median propagation speed c of traveling waves is 1.64 mm/ms (range: 0.75-3.99 mm/ms), 1.63 mm/ms (0.61-4.19 mm/ms), and 2 mm/ms (0.62-5.74 mm/ms) along electrodes 1,

2, and 3 in subject 1; and 1.57 mm/ms (range: 0.85-3.82 mm/ms) and 1.47 mm/ms (0.65-4.1 mm/ms) along electrodes 1 and 2 in subject 2.

In the literature, naturally occurring traveling waves on the global neocortical scale in humans have propagation speeds between of 0.25 to 10 mm/ms (Muller et al., Nature Reviews Neuroscience, 2018 reports 1 to 10 mm/ms; and Zhang et al., Neuron, 2018 reports 0.25 to 3 mm/ms). Our values fall within the same range which is not surprising as they underlie similar physical constraints (frequency range and brain size). Moreover, for TACS-induced traveling waves, the speed can be controlled by changing the frequency of the stimulation currents or electrode locations. We have added the following paragraph to the paper:

(page 14) *The properties of these phenomena [natural travelling waves] vary with respect to the underlying neurophysiological process or task and brain state. In the human brain, traveling wave propagation speed range from 0.25 to 10 mm/ms and its spatial extent varies from few millimeters to dozens of centimeters^{33,34,48}. In our experiments, we generated traveling waves that propagated at speeds of 1.5 to 2 mm/ms. However, multi-electrode TES can directly control the speed and the phase gradient of the electric fields through the choice of stimulation parameters (e.g. frequency, phase, electrode montage).*

2.3) With all the above, the “dissimilarity index” becomes a very superficial and unclear form to quantify the traveling waves. Substituting it by wavelength or spatial frequency will provide much clearer results.

As requested, we have included these additional analyses (see answers 2.1 + 2.2). Irrespective of these new analyses, we have found the dissimilarity index helpful for understanding the data. Further, given that its findings agree with the additional analysis, we prefer to keep it in the manuscript.

2.4) With the dissimilarity index out, you want also drop usage of the normalized electric field. It masks the actual data you present by scaling each contact individually, misleading the reader to think that you have a clear sinusoidal wave along the electrodes. When in reality the amplitude of the wave is varying drastically, and the wavelength will probably also vary significantly in

space. Please remove all the normalized data from the main paper and have the non-normalized data instead.

According to the reviewer's request from the previous round of review, we provided all plots in both normalized and non-normalized forms. We respectfully disagree that the removal of one of these plots can increase the transparency of the findings.

3) **The treatment of the simulations by the authors is very superficial. It is not at all shown that the simulations are “in full agreement with the experimental data”, as claimed in line 416.**

Following the reviewer's suggestions from the comments above, we have conducted an additional analysis using phasor algebra of the electric fields from our FEM simulations. The underlying equations are detailed above (please see question 1.2). These new results support our initial conclusion on the agreement of our experimental and simulation data and provide further evidence for the presence of gradual changes of the electric field phase in the brain. These results are added as Supplementary Figure 10:

Supplementary Figure 10. Phasor analysis of the electric fields from FEM simulations. **a** Electric field phase on the brain surface calculated with phasor algebra. **b** Experimental recordings of electric field phase in subject 1. This figure complements Figure 7 in the main paper.

The following text has been added to the main paper in the results section:

(page 12) *Mathematically, the electric field \vec{E} (including magnitude and phase) arising from phase-shifted TES can be calculated for any brain region leveraging the principles of phasor analysis (see Supplementary Discussion for the mathematical derivation). Performing this analysis shows smooth gradual changes of the electric field phase across the neocortex for stimulation conditions, such as 45°, but not for 0° or 180° (Supplementary Figure 10).*

3.2) Not only because the electric field changes location in time you will have a traveling wave. In addition to that, the movement should be somewhat orderly (p3(x) should be somewhat continuous, not jumping randomly). This was superficially shown for the electrode recordings with the dissimilarity index, even though more analysis is needed (see point 2), but not at all shown for the simulations. A way to do it would be to plot the phase and amplitude at each cortical position. Those can be easily calculated using the phasor sum formulas.

We agree that an orderly, gradual phase shift is an important property of a traveling wave. This is already been shown in the main paper (Figure 4) and Supplementary Figures 5+6. In addition, we further demonstrate this using phasor analysis in the new Supplementary Figure 10 (see question 3 above).

3.1) Simulations show the **norm** of the electric field over time. However, as the authors point out in Line 411 and Line 471, the direction of the electric field is important. So the plots should also probably feature a specific electric field direction, such as the normal direction.

We agree that the direction of the electric field is important. For the directed electric fields in the experimental data, we are already presenting this in Figure 5 (please see below) and animated Movies 2-4. Please note that with the implanted linear array, we are measuring the electric field component along the electrode direction. FEM simulations predict electric fields in all directions (x,y,z) and can complement these experimental results. For the simulation results, the directed electric fields are available in animated Movies 5-7.

Figure 5. Electric fields in the brain over time during TES for a given stimulation condition. The panel depicts the main conditions (0° , 90° , and 180° stimulation phase differences) for subject 1. Arrows indicate the electric field direction, and the color encodes the electric field magnitude. See Supplementary Figure 8 for subject 2 and Movies 2, 3 for more details.

3.2) Working with the normal has other disadvantages. Looking to Eq. R1, it is not clear to me if the norm of the electric fields will have a well-defined phase. In fact, it is easy to show that in some circumstances the norm of the field can be constant.

In addition to the electric field norm, we also present the electric field vector in our simulations (see Movie 5-7).

3.2) As the **normal component** is highly dependent on the local cortical shape, I expect that for the normal component the “traveling wave” will not be present. That is, $p_3(x)$ in Eq. R1 will suffer large discontinuities throughout the cortical surface. Especially when considering humans. Please show some data and discuss the results.

With respect to electric field components (normal, tangential) traveling wave tACS is similar to standard tACS. Both in- and out-flowing currents occur at the cortical surface in addition with strong tangential components. At the moment, it is still subject to ongoing research which one of these components will be the most physiological effective. Thus, we prefer to not extend our analysis to include these components at this point.

Other points:

O.1) Line 26: Which studies need to be re-evaluated? As far as I know, it is only the small subset tACS studies applies “anti-phasic” protocols. And even then, those are limited to 0 and 180 degrees phase differences, as pointed out by the authors in Line 52. The study also shows that there is no traveling wave for 0 degrees and 180 degrees (Line 283). So there will be no studies to be re-evaluated, because this kind of stimulation has never been used. Please remove this sentence from the abstract.

We revisited the abstract and removed the phrase in question. The abstract now concludes that:
“Our results provide a mechanistic understanding of multi-electrode TACS and enable future developments of novel stimulation protocols.”

O.2) Line 65: Please be more clear as to what you mean by “translate linearly into phase differences”. As far as I’m aware, it is assumed that in anti-phasic stimulation the phase is shifted by 180 degrees in the regions underneath the electrodes.

This is exactly what we intended to imply. The line is now clarified:

“The second [previously existing] is that the phase of the electric field in the brain underneath the stimulation electrodes is the same as the phase of the currents passing through these electrodes.”

O.3) Line 175: I think that the term “anti-directional” is inappropriate. Please change it to “bidirectional” or something else with a more clear meaning. .

Following the reviewer’s suggestion, we renamed all instances of this term in the paper to “bidirectional.”

O.4) In the paragraph started in line 269, there are several mentions to the voltage and electric field “data”, and a mention to a non-linear increase and decrease. To which data more precisely are the authors referring to? From the context I infer it would be the rms or the maximum electric field for each phase and in each electrode, but it is not clear. The same is valid elsewhere in the text, such as in the legend of several plots.

We carefully checked all instances of the term “data” in the manuscript and clarified where appropriate if the data represent the voltage or the electric field.

O.5) I could not find the new animations to download. Were they uploaded?

We are apologizing for any technical issues that prevented the download of the new animations. We now re-uploaded them and renamed them as “Movies” in accordance with the journal guidelines.

O.6) It should be made clear that this difference between the in- and anti-phase conditions is highly dependent on the electrode set-up.

We completely agree with this point. Thus, in the Discussion we write:

“The exact features of the electric field will depend both on the stimulation current phases and on the electrode placement, as we show through the additional experiment.”

O.7) Several points in the paper, it is said that the maximum moves. By how much does it move (in mm)? From the raw data in supplementary 7 and 8 I would think it moves little to nothing, and often simply jump between far away contacts. This does not mean you do not have a wave, it just means that the amplitude of the wave changes drastically. But all mentions to the “maxima moving over time” should be rephrased to a less misleading sentence.

As stated in the method section, the distance between adjacent contacts of the recording electrode is 5 mm. Regarding the traveling wave pattern, the electric field maximum orderly moves for the “travelling wave” stimulation condition, most notably for the 45° current phase shift. This movement is especially clear in the normalized data plots (please see Supplementary Figure 8 below). The maximum gradually attends every contact over the stimulation cycle, thus the maximum moves at least at 5 mm fine steps. Considering the non-normalized or raw data, the maximum’s amplitude is indeed higher by factor of two in the most posterior location than anywhere else (even at the most anterior contacts). This is possibly due to the proximity of the posterior contacts to the craniotomy or better alignment of these contacts with the electric field. Besides this most posterior contact, the maximum’s amplitudes are relatively stable across the brain, as well as they are stable in our computer simulation (Figure 7 above). Thus, we would like to keep our initial statement that the maxima is moving over time.

Supplementary Figure 8. *a* Absolute normalized and *b* non-normalized electric field time-courses for subject 1 and electrode 1. The first contact in the electrode corresponds to the most anterior location, and the last contact – to the most posterior location. While for 0° and 180° the maxima across contacts occur at the same time point, they occur at different time points for the 45° condition (= traveling wave). Other electrodes demonstrate a similar pattern. The panel corresponds to Figure 6 in the main paper. Also see Movies 4-7 for a 3d animation.

O.8) Line 409: The authors misrepresent would be the goal of the “in-phase” and “anti-phase” simulation. In the “in-phase” conditions, fields are supposed to be simultaneously excitatory on inhibitory. As the electrodes are positioned in opposing ends of the head and the important field component is the normal component, the electric fields should be “anti-directional”. One more reason why the term “anti-directional” is unfortunate.

As we wrote in the response for O.3, all instances of the term “anti-directional” are now replaced with the suggested term “bidirectional”.

Also, please, notice that we present the goal of the “in-phase” and “anti-phase” stimulation exactly as it was defined in the literature before (Polania et al., Current Biology, 2012) and later re-iterated by other authors (i.e., Violante et al., eLife, 2017; Helfrich et al., PLoS Biology, 2014):

Polania et al., 2012: “We applied tACS at 6 Hz over the left prefrontal and parietal cortices ... with a relative 0° (“synchronized” condition) or 180° (“desynchronized” condition) phase difference”.

Our paper: “an increase in coordination between two areas is assumed when they experience an in-phase stimulation and a disorganization through an anti-phase stimulation.”

O.9) Line 448: No evidence is presented that the proposed stimulation protocol actually does mimic the physiological traveling wave. It might be that, in a more realistic setting, the kind of activity caused by the proposed stimulation protocol is different from the cortical traveling waves.

As we detailed in the answers to questions 2.1 and 2.2, all properties of the traveling wave that we induced in our experiment are within the physiological range (according to Muller et al., Nature Reviews Neuroscience, 2018; Zhang et al., Neuron, 2018; Muller et al., eLife, 2016; and Lozano-Soldevilla & VanRullen, Cell Reports, 2019), including (i) the temporal frequency – 10 Hz (in the human brain – 0.1 to 80 Hz); (ii) the spatial frequency – 1-5 deg/mm (in the human brain – above 1 deg/mm); and (iii) the propagation speed – 1.5-2 mm/ms (in the human brain – 0.25 to 10 mm/ms).

Moreover, please note that all these properties can be directly controlled through the stimulation settings. Thus, using our proposed framework, it is feasible to design traveling wave TES with a wide range of desired properties that will match a specific physiological process.

I have read the paper and comments. In brief, I tend to agree with the position of Reviewer 2. There is a dissonance in what the paper is and how it is presented. To be clear, this is a paper about verification of models, not about discovery of new phenomena. The main findings,

- (i) differing electric field magnitude across stimulation phase conditions;
- (ii) non-linear relationship between transcranial stimulation phase and measured intracranial phase;
- (iii) specific phase configurations can create travelling wave stimulation patterns.

can easily be predicted from electromagnetic theory. That is, what the authors find is not surprising, at least to those trained in electromagnetism or who have been modeling electric fields generated by transcranial stimulation. Software interfaces for current-controlled multichannel systems have been providing visualization tools for tACS for years now using such models, and traveling waves, for example, can easily be simulated using them. All this kind of contradicts the authors' statement "However, phase properties of electric fields arising from multi-electrode tACS have not been considered thus far, either theoretically or experimentally. " But the field is now very interdisciplinary (bringing physics, engineering and neuroscience together), and this sometimes creates gaps across communities.

On the positive side, this is an interesting study and contributes to the field by (positively) contrasting valuable in-vivo data (which is scarce to date) with models in the context of multielectrode tACS.

The authors state:

The current rationale for multi-electrode TACS applications relies on two main assumptions. The first is that the manipulation of phase differences between stimulation electrodes does not significantly change other properties of the generated electric field. Most importantly, this predicts the field's spatial configuration, i.e., the electric field magnitude at different spatial locations. However, direct measurements to substantiate this assumption are lacking. The second assumption is that the phase of the electric field in the brain underneath the stimulation electrodes is the same as the phase of the currents passing through these electrodes. This assumption holds for the two electrode case, where the only possible obstacle for such translation could be capacitive effects introducing additional phase shifts. However, phase properties of electric fields arising from multi-electrode TACS have not been considered thus far, either theoretically or experimentally. Here, we aim to address this gap by measuring electric field magnitudes and phase angles across the brain during three-electrode TACS under varying phase conditions.

I was not aware of the existence of such assumptions. It is true that as far as this reviewer knows, a discussion of the intricacies of multichannel tACS has not been provided in the literature (beyond papers that discuss multifrequency arrangements for interference). There are some interesting aspects, to be sure, and they all follow from a straightforward application of tools normally used in physics or engineering graduate school. In my opinion, this is where the paper should start: there are models that can be applied and then be verified in-vivo.

For example, the general expression for the electric field in multichannel monochromatic tACS is provided by the superposition principle. As usually done in physics (see., e.g., J. D. Jackson, Classical electrodynamics) we can use complex analysis to simplify the algebra, with the convention that the physical electric field is obtained by taking the real part of complex quantities (this is the origin of "phasor analysis"). For example, we can write

$$\mathbf{E}(r, t) = \sum_i \mathbf{E}^i(r) e^{i(\omega t + \phi_i)} = e^{i\omega t} \sum_i \mathbf{E}^i(r) e^{i\phi_i}, \quad (1)$$

where $\mathbf{E}^i(r)$ is the (real) electric field vector at point r due to a bipolar montage with a common reference (any multichannel montage can be written as such a sum, from charge conservation). Because stimulation is assumed here to be monochromatic, the angular frequency ω does not depend on montage. Furthermore, neither ω or ϕ_i depend on the location r (the latter results from the quasistatic approximation¹).

We can also write

$$\sum_i \mathbf{E}^i(r) e^{i\phi_i} = \tilde{\mathbf{E}}(r), \quad (2)$$

where $\tilde{\mathbf{E}}(r)$ is a vector with complex entries encoding the phase and amplitude of each

¹See, e.g., Ruffini et al., IEEE Trans on Neural Sys and Rehab Eng, Vol. 21, No. 3, May 2013

component. Note here, for example, that if all the phases are 0 or 180, $\tilde{\mathbf{E}}(r)$ will be a real vector (and there will be no spatial phase gradients).

Depending on the relative phase and amplitude of components, $\mathbf{E}(r, t)$ will rotate in a plane with a frequency ω ,

$$\mathbf{E}(r, t) = e^{i\omega t} \tilde{\mathbf{E}}(r) = e^{i\omega t} \begin{bmatrix} \tilde{E}_x(r) e^{i\phi_x} \\ \tilde{E}_y(r) e^{i\phi_y} \\ \tilde{E}_z(r) e^{i\phi_z} \end{bmatrix} \sim \begin{bmatrix} \tilde{E}_x(r) \sin(\omega t + \phi_x) \\ \tilde{E}_y(r) \sin(\omega t + \phi_y) \\ \tilde{E}_z(r) \sin(\omega t + \phi_z) \end{bmatrix}. \quad (3)$$

That the field lies in a plane follows from the fact that the equation $0 = \mathbf{n}(r) \cdot \mathbf{E}(r, t)$ or, equivalently, $0 = \mathbf{n}(r) \cdot \tilde{\mathbf{E}}(r)$, can be solved (for both real and imaginary parts in the latter case). To see this, notice $\mathbf{E}(r, t)$ can be broken down as a sum leading to an equation for the $\cos(\omega t)$ and another for $\sin(\omega t)$. More explicitly, we can expand

$$\mathbf{E}(r, t) = \sin(\omega t) \begin{bmatrix} \tilde{E}_x(r) \cos(\phi_x) \\ \tilde{E}_y(r) \cos(\phi_y) \\ \tilde{E}_z(r) \cos(\phi_z) \end{bmatrix} + \cos(\omega t) \begin{bmatrix} \tilde{E}_x(r) \sin(\phi_x) \\ \tilde{E}_y(r) \sin(\phi_y) \\ \tilde{E}_z(r) \sin(\phi_z) \end{bmatrix} \equiv \tilde{\mathbf{E}}_1(r) \sin(\omega t) + \tilde{\mathbf{E}}_2(r) \cos(\omega t) \quad (4)$$

which defines an ellipse in the plane orthogonal to $\mathbf{n}(r) = \tilde{\mathbf{E}}_1(r) \times \tilde{\mathbf{E}}_2(r)$.

If the phases depend on r , the electric field will appear to travel locally (in direction of phase gradient). To see this, we can expand the phases up to first order (if we ignore variations in amplitude, $\phi = \phi_0 + r \cdot \nabla \phi(r)$), or by Fourier expansion (but each wavelength will have its propagation velocity),

$$\tilde{\mathbf{E}} = \int d^3k \tilde{\mathbf{E}}(\omega) e^{i\mathbf{k} \cdot \mathbf{r}}. \quad (5)$$

Implanted electrodes provide access to directed measurement of the field along a direction $\boldsymbol{\lambda}$,

$$M(t) = e^{i\omega t} \boldsymbol{\lambda} \cdot \tilde{\mathbf{E}}(r), \quad (6)$$

which can again be described by a phasor (which can also be 0 if $\boldsymbol{\lambda}$ is parallel to the \mathbf{n} vector above) to create the link between theory and experimental magnitudes and phases.

[.....]

All the above is to show that, based on the underlying physics, a finite element model (which is needed to compute the $\mathbf{E}_i(r)$ in Equation 1) can be used to predict the outcome of measurements and contrast them with experimental data. So, the merits of this paper can be seen as verifying the predictions of modeling with regard to the solution of Poisson's equation and the quasistatic approximation. Some of the conclusions in the current version (and I guess more so in earlier versions) seem to point surprise to things like "non-linear" behavior which, as Reviewer 2 points out, are fully expected from the underlying physics. The value of this paper really is, as the authors state in the discussion, that they

... performed electric field modeling and phasor analysis of the main experimental conditions. These models describe all directions of the electric fields in the brain and are in full agreement with the experimental results.

together with figures 10 and 7. This should be the conceptual center of the paper.

The results of this work do not really provide a mechanistic understanding of the effects to tACS, as the authors claim in the abstract. Rather, they confirm what theory predicts with regard to “passive” biophysical effects of brain stimulation (non-physiological)—which is great all by itself. This theory is essentially a direct consequence of Maxwell’s equations. The work provides a validation of models of the electric field that researchers are now using now the time. They represent the first step in developing a mechanistic understanding of effects (the next one is linking fields to physiology, which the paper does not address).

In summary, the paper is valuable but should be rewritten starting from a clear theoretical electromagnetism foundation to then make predictions using FEM and then continuing to experimental validation. Theory and modeling should be the driver of the paper, not an afterthought. I think the authors, with the help of reviewers, have done most of the work already.

We want to thank the reviewer for his constructive comments regarding our manuscript. Based on the comments we have made the following changes to the manuscript (marked in the main paper as tracked changes):

1. We expanded the introduction on how electromagnetic theory can be used to describe electric fields from multielectrode TACS. Specific changes to the text as suggested by the reviewer are outlined below:
 - a) We changed the sentence “*However, phase properties of electric fields arising from multi-electrode TACS have not been considered thus far, either theoretically or experimentally*” to:

“However, fundamental electromagnetic theory predicts, based on the superposition principle, that a spatially varying phase gradient in the electric field can arise under certain conditions from multi-electrode TACS (see Supplementary Discussion). While such behavior can be expected theoretically, it has not been demonstrated experimentally using *in vivo* recordings in the brain.”
 - b) We changed the sentence “*Our results provide a mechanistic understanding of multi-electrode TACS and enable future developments of novel stimulation protocols*” to: “Our results provide a mechanistic understanding of the biophysics of multi-electrode TACS and enable future developments of novel stimulation protocols.”
2. We expanded the Supplementary Discussion to better explain the theory of how a phase gradient can arise in multielectrode TACS:

“[...] Thus, phasor analysis provides a way for the precise characterization of phase relationships of the electric field during multi-electrode TACS with any arbitrary configuration of input currents. From Supplementary Equation 9 it follows that the resulting phase φ_E^x will vary spatially (except when $\varphi_1, \varphi_2 = 0$ or 180 degrees). The resulting phase angle will depend on the ratio of E_1^x to E_2^x , with the dominant component determining its value (see Supplementary Figure 2). As the ratio of E_1^x and E_2^x will change with the location in the brain since these electric fields arise from two different montages, the resulting phase angle φ_E^x will vary spatially (see Supplementary Figure 2). The experimentally measured electric field

E_{meas} captures the component of the TACS electric field \mathbf{E} along the direction \mathbf{d} of the implanted electrode array:

$$E_{\text{meas}} = \mathbf{d} \cdot \mathbf{E} \quad (11)$$

Considerations, as conducted here for the x-component, can be performed equivalently for y and z components or any arbitrary direction such as the direction of the implanted electrode array. The phasor formulation thus facilitates efforts to simulate multielectrode TACS electric fields using the finite element method which can be compared to the measured electric field (see example in Supplementary Figure 11).

Supplementary Figure 2. Illustration of the phasor addition principle. A phasor \mathbf{P}_1 with $\varphi_1 = 0$ degrees and a phasor \mathbf{P}_2 with $\varphi_2 = 90$ degrees are added at different brain locations. The resulting phasor \mathbf{P}_3 will exhibit a phase angle φ_3 according to Supplementary Equation 9 which will vary across different brain regions.”